# VECTOR QUANTIZATION BY DISTRIBUTION MATCHING

## ABSTRACT

The success of autoregressive models largely depends on the effectiveness of vector quantization, a technique that compresses and discretizes continuous features by mapping them to the nearest code vectors within a learnable codebook. Two critical issues in existing vector quantization methods are training instability and codebook collapse. Training instability arises from the gradient gap during both forward and backward gradient propagation, especially in the presence of significant quantization errors, while codebook collapse occurs when only a small subset of code vectors are utilized during training. A closer examination of these issues reveals that they are primarily driven by a mismatch between the distributions of the features and code vectors, leading to unrepresentative code vectors and significant data information loss during compression. To address this, we employ the Wasserstein distance to align these two distributions, achieving near 100% codebook utilization and significantly reducing the quantization error. Both empirical and theoretical analyses validate the effectiveness of the proposed approach.

## 1 INTRODUCTION

Autoregressive models have experienced a resurgence in visual generative models (Razavi et al., 2019; Esser et al., 2021; Chang et al., 2022; Lee et al., 2022; Tian et al., 2024). This revival is marked by the superior quality of images generated through autoregressive methods, which have now surpassed those produced by diffusion-based approaches (Ho et al., 2020; Rombach et al., 2022; Sun et al., 2024; Tian et al., 2024; Ma et al., 2024). The success of autoregressive visual generative models hinges on the effectiveness of vector quantization (VQ) (van den Oord et al., 2017), a technique that compresses and discretizes continuous features by mapping them to the nearest code vectors within a learnable codebook. However, VQ continues to face two major challenges: training instability and codebook collapse.

The first issue arises from the non-differentiability of VQ, which prevents direct gradient backpropagation from quantized features to their continuous counterparts (see more in Section 2.1), thereby hindering model optimization. VQ-VAE (van den Oord et al., 2017) addresses this challenge using a straightforward approach by employing the straight-through estimator, which allows gradients to be copied from quantized to continuous features. However, this method introduces a significant gradient gap in both forward and backward gradient propagation, particularly in the presence of large quantization errors, leading to unstable training (Lee et al., 2022; Zhu et al., 2024a).

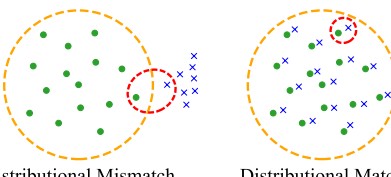

Distributional Mismatch    Distributional Match

Figure 1: The symbols $\cdot$ and $\times$ represent the feature and code vectors, respectively. The left figure illustrates the distributional mismatch between the feature and code vectors, while the right figure visualizes their distributional match.

The latter issue occurs when only a small subset of code vectors are updated during optimization, leaving the majority unused and unoptimized (Zheng & Vedaldi, 2023). Although various methods have been proposed to tackle this issue, they often fail to fully leverage the expressive capacity of the codebook due to low utilization of code vectors, particularly when the codebook size is large (Dhariwal et al., 2020; Takida et al., 2022; Yu et al., 2022; Lee et al., 2022; Zheng & Vedaldi, 2023). Consequently, the effectiveness of VQ is markedly compromised.

In this paper, we examine these issues by investigating the distributions of the features and code vectors. To illustrate the idea, Figure 1 presents two extreme scenarios: the left panel depicts a significant mismatch between the two distributions, while the right panel shows a match. In the left panel, all features are mapped to a single codeword, resulting in large quantization errors and minimal codebook utilization. In contrast, the right panel demonstrates that a distributional match leads to negligible quantization error and near 100% codebook utilization. This suggests aligning these two distributions in VQ could potentially address the issues of training instability and codebook collapse.

**Contributions** To investigate the idea above, we first introduce three principled criteria that a VQ method should optimize. Guided by this criterion triple, we conduct qualitative and quantitative analyses, demonstrating that aligning the distributions of the feature and code vectors results in near 100% codebook utilization and minimal quantization error. Additionally, our theoretical analysis underscores the importance of distribution matching for vector quantization. To achieve this alignment, we employ the quadratic Wasserstein distance which has a closed-form representation under a Gaussian hypothesis. Our approach effectively mitigates both training instability and codebook collapse, thereby enhancing image reconstruction performance in visual generative tasks.

## 2 UNDERSTANDING DISTRIBUTION MATCHING

This section introduces a novel distributional perspective for VQ and investigates the effects of distribution matching. We begin with an overview of VQ and then identify three principled criteria. Utilizing this criterion triple, we conduct qualitative and quantitative analyses. Our empirical findings demonstrate that distribution matching yields the optimal criterion triple, a conclusion further supported by our theoretical analysis.

### 2.1 AN OVERVIEW OF VECTOR QUANTIZATION

The seminar work PixelCNN (van den Oord et al., 2016) achieved autoregressive visual generation by treating image pixels as sequential tokens. However, this approach involves long token sequences, resulting in significant time and computational costs, particularly during the sequential generation of pixels. To reduce these expenses, VQ (van den Oord et al., 2017) was introduced. VQ alleviates these costs by learning image tokens in much shorter sequences within the latent space.

Figure 2 illustrates the classic VQ process (van den Oord et al., 2017), which consists of an encoder $E(\cdot)$, a decoder $D(\cdot)$, and an updatable codebook $\{\mathbf{e}_k\}_{k=1}^K \in \mathbb{R}^{K \times d}$ containing a finite set of code vectors. Here, $K$ represents the size of the codebook, and $d$ denotes the dimension of the code vectors. Given an image $\boldsymbol{x} \in \mathbb{R}^{H \times W \times 3}$, the goal is to derive a spatial collection of codeword IDs $r \in \mathbb{N}^{h \times w}$ as image tokens. This is achieved by passing

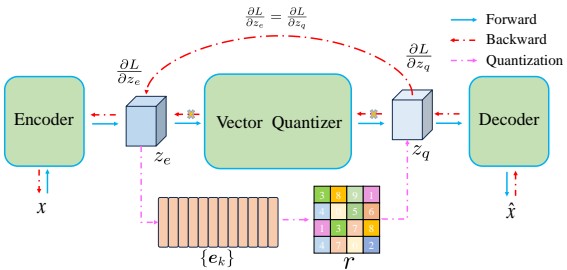

Figure 2: An illustration of VQ.

the image through the encoder to obtain $\boldsymbol{z}_e = E(\boldsymbol{x}) \in \mathbb{R}^{h \times w \times d}$, followed by a spatial-wise quantizer $\mathcal{Q}(\cdot)$ that maps each spatial feature $\boldsymbol{z}_e^{ij}$ to its nearest code vector $\boldsymbol{e}_k$:

$$r^{ij} = \arg\min_k \|\boldsymbol{z}_e^{ij} - \boldsymbol{e}_k\|_2^2. \tag{1}$$

These tokens are used to retrieve the codebook entries $\boldsymbol{z}_q^{ij} = \mathcal{Q}(\boldsymbol{z}_e^{ij}) = \boldsymbol{e}_{r^{ij}}$, which then pass through the decoder to reconstruct the image as $\widehat{\boldsymbol{x}} = D(\boldsymbol{z}_q)$. VQ significantly reduces sequence length during image tokenization, since $h \times w$ (the token length) is much smaller than $H \times W$. However, despite its success in high-fidelity image synthesis (van den Oord et al., 2017; Razavi et al., 2019; Esser et al., 2021), VQ faces two key challenges: training instability and codebook collapse.

**Training Instability** This issue occurs because during backpropagation, the gradient of $\boldsymbol{z}_q$ cannot flow directly to $\boldsymbol{z}_e$ due to the non-differentiable function $\mathcal{Q}$. To optimize the encoder's network parameters through backpropagation, VQ-VAE (van den Oord et al., 2017) employs the straight-through estimator (STE) (Bengio et al., 2013), which copies gradients directly from $\boldsymbol{z}_q$ to $\boldsymbol{z}_e$. However,

this approach carries significant risks—especially when $z_q$ and $z_e$ are far apart. In these cases, the gradient gap between the representations can grow substantially, destabilizing the training process.

To address this instability, RQ-VAE (Lee et al., 2022) introduces residual quantization to minimize the distance between $z_q$ and $z_e$, thereby reducing the gradient gap. VAR (Tian et al., 2024) builds on this approach by implementing residual quantization from a multi-scale perspective. In this paper, we tackle the training instability challenge from a distributional viewpoint.

**Codebook Collapse**  Codebook collapse occurs when only a small subset of code vectors receives optimization-useful gradients, while most remain unrepresentative and unupdated (Dhariwal et al., 2020; Takida et al., 2022; Yu et al., 2022; Lee et al., 2022; Zheng & Vedaldi, 2023). Researchers have proposed various solutions to this problem, such as improved codebook initialization (Zhu et al., 2024a), reinitialization strategies (Dhariwal et al., 2020; Williams et al., 2020), and classical clustering algorithms like $k$-means (Bradley & Fayyad, 1998) and $k$-means++(Arthur & Vassilvitskii, 2007) for codebook optimization (Razavi et al., 2019; Zheng & Vedaldi, 2023). Beyond these deterministic approaches that select the best-matching token, researchers have also explored stochastic quantization strategies (Zhang et al., 2023; Ramesh et al., 2021; Takida et al., 2022).

However, these methods still fail to fully utilize the codebook's expressive power due to low utilization rates, particularly with large codebook sizes $K$ (Zheng & Vedaldi, 2023; Mentzer et al., 2024). In this paper, we address this limitation by implementing distribution matching between feature vectors and code vectors.

## 2.2 EVALUATION CRITERIA

We assume that all continuous feature vectors[1] $z_i$ follow a distribution $\mathcal{P}_A$, while all code vectors $\mathbf{e}_k$ follow a distribution $\mathcal{P}_B$[2]. We aim to determine the optimal codebook distribution $\mathcal{P}_B$ for a given feature distribution $\mathcal{P}_A$. Our optimality criteria are based on three key aspects of VQ: quantization error, codebook utilization, and codebook perplexity.

Given a set of feature vectors $\{z_i\}_{i=1}^N$ and code vectors $\{e_k\}_{k=1}^K$, vector quantization involves finding the nearest, and thus most representative, code vector for each feature vector:

$$z_i' = \arg\min_{e \in \{e_k\}} \|z_i - e\|.$$

The original feature vector $z_i$ is then quantized to $z_i'$. Below, we introduce three key criteria to evaluate this process.

**Criterion 1** (Quantization Error). *The quantization error measures the average distortion introduced by VQ and is defined as*

$$\mathcal{E}(\{e_k\}; \{z_i\}) = \frac{1}{N} \sum_i \|z_i - z_i'\|^2.$$

A smaller $\mathcal{E}$ signifies a more accurate quantization of the original feature vectors, resulting in a smaller gradient gap between $z_i$ and $z_i'$. Consequently, a small value of $\mathcal{E}$ suggests that the issue of training instability can be effectively mitigated.

**Criterion 2** (Codebook Utilization Rate). *The codebook utilization rate measures the proportion of code vectors used in VQ and is defined as*

$$\mathcal{U}(\{e_k\}; \{z_i\}) = \frac{1}{N} \sum_{i=1}^N \mathbf{1}(e_k = z_i' \text{ for some } i).$$

A higher value of $\mathcal{U}$ reduces the risk of codebook collapse. Ideally, $\mathcal{U}$ should reach 100%, indicating that all code vectors are actively utilized. As discussed in Appendix H.1, $\mathcal{U}$ can only measure the *completeness* of codebook utilization; it does not suffice to evaluate the degree of codebook collapse. This motivates us to introduce the codebook perplexity criterion.

---

[1]For simplicity. the spatial feature, denoted as $z_e^{ij}$ in Section 2.1, is written as $z_i$ in Section 2.2.

[2]For clarity in notation, we use "codebook distribution" to refer to the code vector distribution and "feature distribution" to refer to the feature vector distribution.

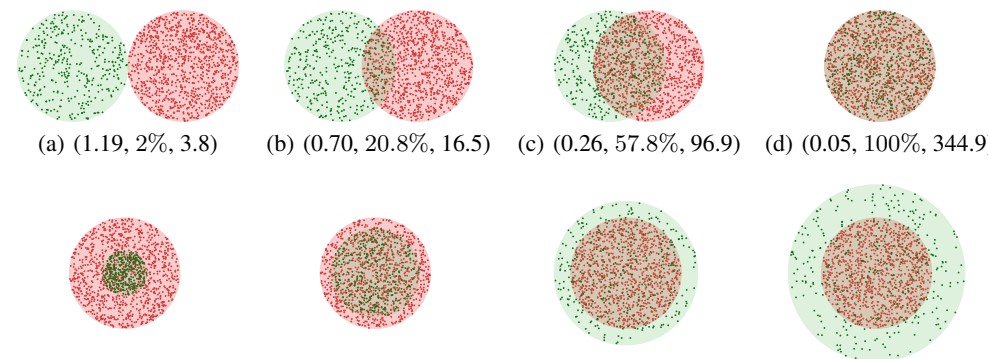

(a) $(1.19, 2\%, 3.8)$    (b) $(0.70, 20.8\%, 16.5)$    (c) $(0.26, 57.8\%, 96.9)$    (d) $(0.05, 100\%, 344.9)$

(e) $(0.36, 93.3\%, 63.2)$    (f) $(0.10, 99.8\%, 250.5)$    (g) $(0.07, 61.3\%, 199.7)$    (h) $(0.08, 45.3\%, 151.5)$

Figure 3: Qualitative analyses of the criterion triple $(\mathcal{E}, \mathcal{U}, \mathcal{C})$: The red and green disks represent the uniform distributions of feature vectors and code vectors, respectively.

**Criterion 3** (Codebook Perplexity). *The codebook perplexity measures the uniformity of codebook utilization in VQ and is defined as*

$$\mathcal{C}(\{\boldsymbol{e}_k\}; \{\boldsymbol{z}_i\}) = \exp(-\sum_{k=1}^{K} p_k \log p_k), \quad p_k := p_k(\boldsymbol{e}_k; \{\boldsymbol{z}_i\}) = \frac{1}{N} \sum_{i=1}^{N} \mathbf{1}(\boldsymbol{z}_i' = \boldsymbol{e}_k).$$

A higher value of $\mathcal{C}$ indicates that code vectors are more uniformly selected in the VQ process. Ideally, $\mathcal{C}$ reaches its maximum at $\mathcal{C}_0 = \exp(-\sum_{k=1}^{K} \frac{1}{K} \log \frac{1}{K})$ when code vectors are completely uniformly utilized. Therefore, as a complementary measure to Criterion 2, the combination of $\mathcal{U}$ and $\mathcal{C}$ can effectively evaluate the degree of codebook collapse.

We refer to $(\mathcal{E}, \mathcal{U}, \mathcal{C})$ as the criterion triple. When comparing extreme cases of distributional match and mismatch shown in Figure 1, we find that distributional matching significantly outperforms mismatching across all three criteria. Using this criterion triple, we present detailed analyses that demonstrate the advantages of distribution matching.

## 2.3 THE EFFECTS OF DISTRIBUTION MATCHING

**A Prototypical Study**    We begin by conducting a simple synthetic experiment to provide intuitive insights[3]. Specifically, we assume that the distributions $\mathcal{P}_A$ and $\mathcal{P}_B$ are uniform distributions confined within two distinct disks, as depicted in Figure 3. We then sample a set of feature vectors $\{\boldsymbol{z}_i\}_{i=1}^{N}$ uniformly from the red disk, and a set of code vectors $\{\boldsymbol{e}_k\}_{k=1}^{K}$ uniformly from the green circle. The criterion triple $(\mathcal{E}, \mathcal{U}, \mathcal{C})$ is then calculated based on the definitions in Criteria 1 to 3.

We examine two cases. The first involves two disks with identical radii but different centers. As shown in Figures 3(a) to 3(d), when the centers of the disks move closer together, aligning the two distributions, the criterion triple improves toward optimal values. Specifically, $\mathcal{E}$ decreases from 1.19 to 0.05, $\mathcal{U}$ rises from $2\%$ to $100\%$, and $\mathcal{C}$ increases from 3.8 to 344.9.

The second case shows two distributions with identical centers but different radii. When the codebook distribution's support lies within the feature distribution's support (as shown in Figures 3(e) and 3(f)), it results in a notably larger $\mathcal{E}$, slightly lower $\mathcal{U}$, and significantly smaller $\mathcal{C}$ compared to the aligned distributions shown in Figure 3(d). Conversely, when the codebook distribution's support extends beyond the feature distribution's support, $\mathcal{E}$ shows a modest increase while both $\mathcal{U}$ and $\mathcal{C}$ decrease significantly, as illustrated in Figures 3(g) and 3(h). We provide detailed explanations of these experimental findings in Appendix H.2.

**More Quantitative Analyses**    To further elucidate the benefits of the distributional matching, we conduct more quantitative analyses centered around the criterion triple $(\mathcal{E}, \mathcal{U}, \mathcal{C})$. We begin by assuming that the distributions $\mathcal{P}_A$ and $\mathcal{P}_B$ are Gaussian[4]. We generate a set of feature vectors

---

[3]The experimental details for all analyses in Section 2.3 are provided in Appendix E.1.

[4]Alternative distribution choices, such as the uniform distribution discussed in Appendix C, are also possible for $\mathcal{P}_A$ and $\mathcal{P}_B$.

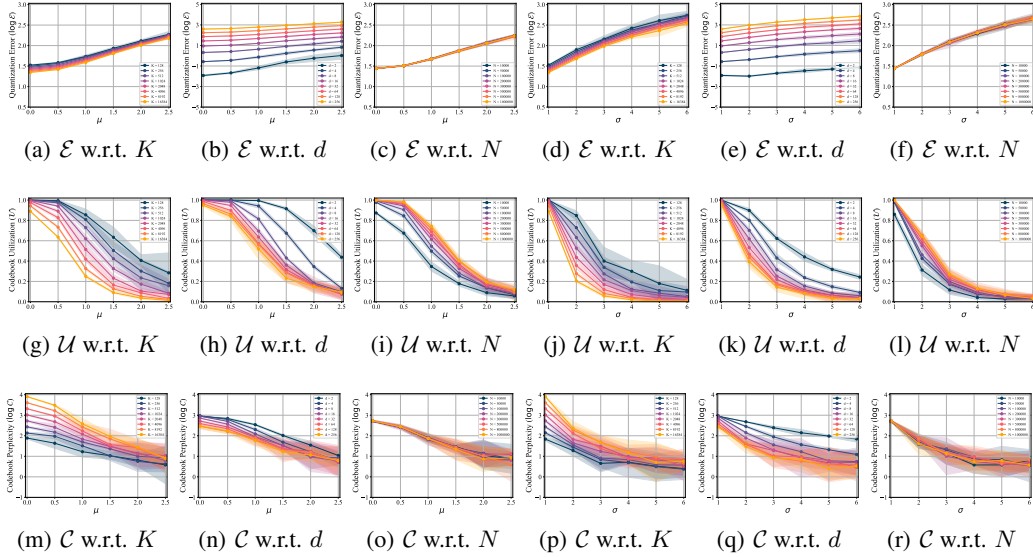

(a) $\mathcal{E}$ w.r.t. $K$    (b) $\mathcal{E}$ w.r.t. $d$    (c) $\mathcal{E}$ w.r.t. $N$    (d) $\mathcal{E}$ w.r.t. $K$    (e) $\mathcal{E}$ w.r.t. $d$    (f) $\mathcal{E}$ w.r.t. $N$

(g) $\mathcal{U}$ w.r.t. $K$    (h) $\mathcal{U}$ w.r.t. $d$    (i) $\mathcal{U}$ w.r.t. $N$    (j) $\mathcal{U}$ w.r.t. $K$    (k) $\mathcal{U}$ w.r.t. $d$    (l) $\mathcal{U}$ w.r.t. $N$

(m) $\mathcal{C}$ w.r.t. $K$    (n) $\mathcal{C}$ w.r.t. $d$    (o) $\mathcal{C}$ w.r.t. $N$    (p) $\mathcal{C}$ w.r.t. $K$    (q) $\mathcal{C}$ w.r.t. $d$    (r) $\mathcal{C}$ w.r.t. $N$

Figure 4: Quantitative analyses of the criterion triple when $\mathcal{P}_A$ and $\mathcal{P}_B$ are Gaussian distributions.

$\{z_i\}_{i=1}^N$ from $\mathcal{N}(\mathbf{0}_d, I_d)$ and a set of code vectors $\{e_k\}_{k=1}^K$ from $\mathcal{N}(\mu \cdot \mathbf{1}_d, I_d)^5$, with $\mu$ varying within $\{0.0, 0.5, 1.0, 1.5, 2.0, 2.5\}$. The criterion triple results are presented in Figures 4(a) to 4(c), Figures 4(g) to 4(i), and Figures 4(m) to 4(o). Across all tested configurations of $K, d, N$, we consistently observe that when $\mu = 0$ — indicating identical distributions between $\mathcal{P}_A$ and $\mathcal{P}_B$ — the criterion triple achieves the lowest $\mathcal{E}$, highest $\mathcal{U}$, and largest $\mathcal{C}$. This empirical evidence reinforces the effectiveness of aligning feature and codebook distributions in VQ.

Additionally, we conduct experiments to investigate the criterion triple by varying the covariance matrix. We sample a set of feature vectors $\{z_i\}_{i=1}^N$ from the distribution $\mathcal{N}(\mathbf{0}, I_d)$ and a corresponding set of code vectors $\{e_k\}_{k=1}^K$ from $\mathcal{N}(\mathbf{0}, \sigma^2 I_d)$, where $\sigma$ is selected from $\{1, 2, 3, 4, 5, 6\}$. The results for the criterion triple are shown in Figures 4(d) to 4(f), Figures 4(j) to 4(l), and Figures 4(p) to 4(r). When $\sigma = 1$, indicating identical distributions between $\mathcal{P}_A$ and $\mathcal{P}_B$, all three evaluation criteria reach their optimal values: the lowest $\mathcal{E}$, highest $\mathcal{U}$, and largest $\mathcal{C}$ across all tested values of $K, d, N$. This result corroborates our earlier findings.

## 2.4 THEORETICAL ANALYSES

In this section, we provide theoretical evidence to support our empirical observations. Let the code vectors $\{e_k\}_{k=1}^K$ and feature vectors $\{z_i\}_{i=1}^N$ be independently and identically drawn from $\mathcal{P}_B$ and $\mathcal{P}_A$, respectively. We say a codebook $\{e_k\}_{k=1}^K$ attains full utilization asymptotically with respect to $\{z_i\}_{i=1}^N$ if the codebook utilization rate $\mathcal{U}(\{e_k\}_{k=1}^K; \{z_i\}_{i=1}^N)$ tends to 1 in probability as $N$ approaches infinity:

$$\mathcal{U}(\{e_k\}_{k=1}^K; \{z_k\}_{i=1}^N) \xrightarrow{p} 1, \quad \text{as } N \to \infty.$$

For the codebook distribution $\mathcal{P}_B$, we say it attains full utilization asymptotically with respect to $\mathcal{P}_A$ if, with probability 1, the randomly generated codebook $\{e_k\}_{k=1}^K$ achieves full utilization asymptotically.

Additionally, a codebook distribution $\mathcal{P}_B$ is said to have vanishing quantization error asymptotically with respect to a domain $\Omega \subseteq \mathbb{R}^d$ if the quantization error over all data of size $N$ tends to zero in probability as $K$ approaches infinity:

$$\sup_{\{z_i\} \subseteq \Omega} \mathcal{E}(\{e_k\}_{k=1}^K; \{z_i\}_{i=1}^N) \xrightarrow{p} 0, \quad \text{as } K \to \infty. \tag{2}$$

Our first theorem shows that $\overline{\text{supp}(\mathcal{P}_A)} = \overline{\text{supp}(\mathcal{P}_B)}$ is sufficient and necessary for the codebook distribution $\mathcal{P}_B$ to attain both full utilization and vanishing quantization error asymptotically. For simplicity, $\mathcal{P}_A$ is assumed to have a density function $f_A$ with bounded support $\Omega \subseteq \mathbb{R}^d$.

---

[5]$\mathbf{1}_d$ represents the vector of all ones.

**Theorem 1.** *Assume $\Omega = \text{supp}(\mathcal{P}_A)$ is a bounded open area. The codebook distribution $\mathcal{P}_B$ attains full utilization and vanishing quantization error asymptotically if and only if $\overline{\text{supp}(\mathcal{P}_B)} = \overline{\text{supp}(\mathcal{P}_A)}$, where $\overline{\mathcal{S}}$ denotes the closure of the set $\mathcal{S}$.*

Theorem 1 establishes the optimal support of the codebook distribution. The boundedness of $\Omega$ is required as we consider the worst case quantization error in equation 2. In real applications, when $\mathcal{P}_A$ follows an absolutely continuous distribution over an unbounded domain, then $\{z_i\}_{i=1}^N$ generated from $\mathcal{P}_A$ will be bounded with high probability. Thus, Theorem 1 also provides theoretical insights for a target distribution $\mathcal{P}_A$ with an unbounded domain.

Besides the optimal support, we also determine the optimal density of the codebook distribution by invoking existing results characterizing asymptotic optimal quantizers (Graf & Luschgy, 2000). Specifically, we consider the case where $N$ approaches to infinity and define the expected quantization error of a codebook $\{e_k\}$ with respect to $\mathcal{P}_A$ as

$$\mathcal{E}(\{e_k\}_{k=1}^K; \mathcal{P}_A) = \mathbb{E}_{z \sim \mathcal{P}_A} \min_{e \in \{e_k\}} \|z - e\|^2.$$

A codebook $\{e_k^*\}_{k=1}^K$ is called the set of optimal centers for $\mathcal{P}_A$ if it achieves the minimal quantization error:

$$\mathcal{E}(\{e_k^*\}_{k=1}^K; \mathcal{P}_A) = \min_{\{e_k\}_{k=1}^K} \mathcal{E}(\{e_k\}_{k=1}^K; \mathcal{P}_A).$$

Theorem 2 demonstrates that, under weak regularity conditions, the empirical measure of the optimal centers for $\mathcal{P}_A$ converges in distribution to a fixed distribution determined by $\mathcal{P}_A$. Notably, we do not assume a bounded domain in the following theorem.

**Theorem 2** (Theorem 7.5, Graf & Luschgy (2000)). *Suppose $Z \sim \mathcal{P}_A$ is absolutely continuous with respect to the Lesbegue measure in $\mathbb{R}^d$ and $\mathbb{E}\|Z\|^{2+\delta} < \infty$ for some $\delta > 0$. Then the empirical measure of the optimal centers for $\mathcal{P}_A$,*

$$\frac{1}{K} \sum_{k=1}^K \delta_{e_k^*},$$

*converges weakly to a fixed distribution $\mathcal{P}_A^*$, whose density function $f_A^*$ is proportional to $f_A^{d/(d+2)}$.*

Theorem 2 implies that $\mathcal{P}_B = \mathcal{P}_A^*$ is the optimal codebook distribution in the asymptotic regime as $K$ approaches infinity. In high-dimensional spaces with large $d$, this optimal distribution $\mathcal{P}_B = \mathcal{P}_A^*$ closely approximates $\mathcal{P}_A$. This further motivates us to align the codebook distribution $\mathcal{P}_B$ with the feature distribution $\mathcal{P}_A$.

## 3 DISTRIBUTION MATCHING VIA WASSERSTEIN DISTANCE

In this section, we propose using the quadratic Wasserstein distance to achieve a distributional match between $\mathcal{P}_A$ and $\mathcal{P}_B$.

### 3.1 WASSERSTEIN DISTANCE

We assume a Gaussian hypothesis for the distributions of both the feature and code vectors. For computational efficiency, we employ the quadratic Wasserstein distance, as defined in Appendix B, to align these two distributions. Although other statistical distances, such as the Kullback-Leibler divergence (Kingma & Welling, 2014; Ho et al., 2020), are viable alternatives, they lack simple closed-form representations, making them computationally expensive. The following lemma provides the closed-form representation for the quadratic Wasserstein distance between two Gaussian distributions.

**Lemma 3** ((Olkin & Pukelsheim, 1982)). *The quadratic Wasserstein distance between $\mathcal{N}(\boldsymbol{\mu}_1, \boldsymbol{\Sigma}_1)$ and $\mathcal{N}(\boldsymbol{\mu}_2, \boldsymbol{\Sigma}_2)$ is*

$$\sqrt{\|\boldsymbol{\mu}_1 - \boldsymbol{\mu}_2\|^2 + \text{tr}(\boldsymbol{\Sigma}_1) + \text{tr}(\boldsymbol{\Sigma}_2) - 2\,\text{tr}((\boldsymbol{\Sigma}_1^{\frac{1}{2}} \boldsymbol{\Sigma}_2 \boldsymbol{\Sigma}_1^{\frac{1}{2}})^{\frac{1}{2}})}. \tag{3}$$

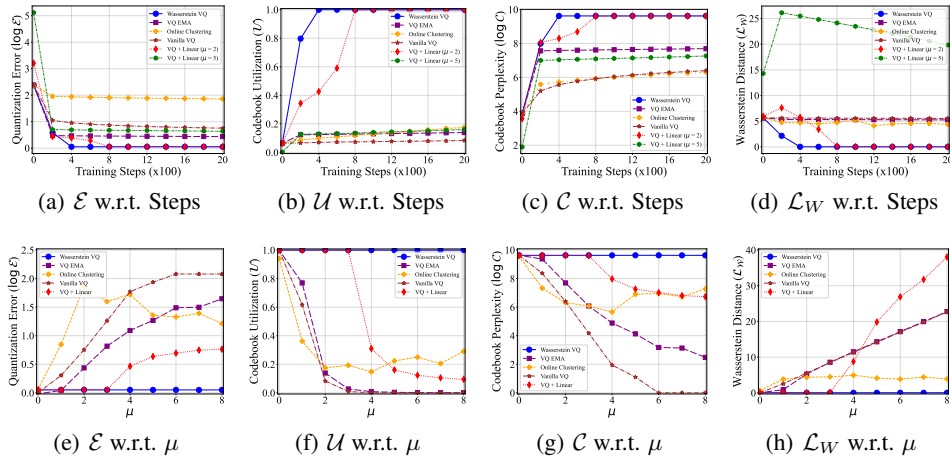

Figure 5: The performance metrics $(\mathcal{E}, \mathcal{U}, \mathcal{C})$ for various VQ approaches.

The lemma above indicates that the quadratic Wasserstein distance can be easily computed using the population means and covariance matrices. In practice, we estimate these population quantities, $\boldsymbol{\mu}_1, \boldsymbol{\mu}_2, \boldsymbol{\Sigma}_1$, and $\boldsymbol{\Sigma}_2$, with their sample counterparts: $\widehat{\boldsymbol{\mu}}_1, \widehat{\boldsymbol{\mu}}_2, \widehat{\boldsymbol{\Sigma}}_1$, and $\widehat{\boldsymbol{\Sigma}}_2$. The empirical quadratic Wasserstein distance is then used as the optimization objective to align the distributions of the feature and code vectors:

$$\mathcal{L}_{\mathcal{W}} = \sqrt{\|\widehat{\boldsymbol{\mu}}_1 - \widehat{\boldsymbol{\mu}}_2\|^2 + \mathrm{tr}(\widehat{\boldsymbol{\Sigma}}_1) + \mathrm{tr}(\widehat{\boldsymbol{\Sigma}}_2) - 2\,\mathrm{tr}((\widehat{\boldsymbol{\Sigma}}_1^{\frac{1}{2}} \widehat{\boldsymbol{\Sigma}}_2 \widehat{\boldsymbol{\Sigma}}_1^{\frac{1}{2}})^{\frac{1}{2}})}. \tag{4}$$

A smaller value of $\mathcal{L}_{\mathcal{W}}$ indicates stronger alignment between the feature distribution $\mathcal{P}_A$ and the codeword distribution $\mathcal{P}_B$. We refer to the VQ algorithm that employs $\mathcal{L}_{\mathcal{W}}$ as *Wasserstein VQ*.

## 3.2 ADVANTAGES OVER OTHER VQ METHODS

In this section, we compare our proposed *Wasserstein VQ* with other VQ algorithms in a simple, atomic experimental setting while examining existing VQ methods from a distributional matching perspective. Specifically, we fix the feature distributions for all VQ methods to the same Gaussian distributions by sampling feature vectors $\boldsymbol{z}_i \sim \mathcal{N}(\mu \cdot \mathbf{1}_d, I_d)$. While feature distributions are typically complex and dynamic in practical training scenarios, this simplified setting still yields valuable insights. We also initialize the codebook distribution as the standard Gaussian distribution across all VQ methods by generating a set of code vectors $\{\boldsymbol{e}_k\}_{k=1}^K \sim \mathcal{N}(\mathbf{0}_d, I_d)$.

Our baseline includes `Vanilla VQ` (van den Oord et al., 2017), `VQ EMA` (which uses exponential moving average updates and is also known as $k$-means in VQ-VAE-2) (Razavi et al., 2019), `Online Clustering` (which employs $k$-means++ in CVQ-VAE) (Zheng & Vedaldi, 2023), and `VQ+Linear` (which incorporates a linear layer projection for code vectors) (Zhu et al., 2024a;b). In all VQ algorithms, we treat sampled code vectors as trainable parameters and optimize them using these algorithms. For detailed experimental specifications, see Appendix E.3.

As illustrated in Figure 5, we evaluate five distinct VQ methods using the criterion triple $(\mathcal{E}, \mathcal{U}, \mathcal{C})$. In Figures 5(a)-5(d), we set $\mu = 2$ except `VQ+Linear`. Our results show that `Vanilla VQ`, `VQ EMA`, and `Online Clustering` exhibit poor VQ performance and substantial Wasserstein distance. This suggests that `Vanilla VQ` and methods based on $k$-means and $k$-means++ are ineffective VQ strategies, since they fail to align the distributions of features and codebooks.

Conversely, `VQ+Linear` achieves both superior VQ performance and a significantly reduced Wasserstein distance, approaching zero at $\mu = 2$. This demonstrates that `VQ+Linear`'s exceptional performance stems from its successful distribution alignment. However, with a large initial distribution gap between codebook and features (at $\mu = 5$), `VQ+Linear` struggles to minimize the distributional distance, resulting in poor VQ performance. This limitation becomes more apparent when varying $\mu$ from 0 to 8, as depicted in Figures 5(e)-5(h), `VQ+Linear` becomes ineffective at $\mu = 4$. These results indicate that `VQ+Linear` remains heavily dependent on codebook initialization. In comparison, our *Wasserstein VQ* algorithm consistently performs best with relatively large values of $\mu$ (e.g., $\mu \geq 4$), thanks to its explicit distributional matching regularization that eliminates reliance on codebook initialization to prevent distributional mismatch.

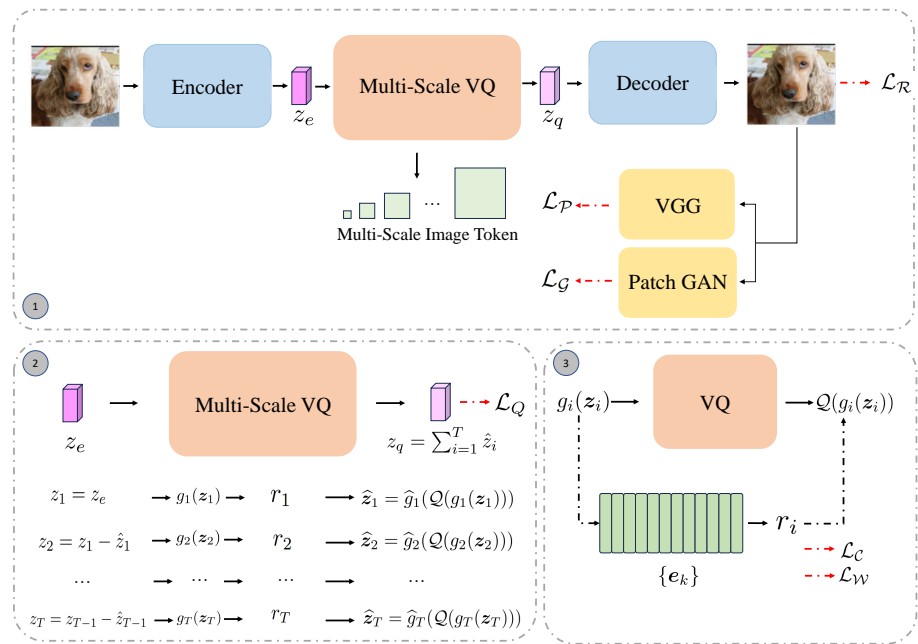

Figure 6: The architecture integrates an encoder-decoder network with a multi-scale VQ module. Block 1 features the standard VQGAN framework (Esser et al., 2021), which includes reconstruction loss $\mathcal{L}_R$, VGG-based perceptual loss $\mathcal{L}_P$, and GAN loss $\mathcal{L}_G$. Block 2 implements the multi-scale VQ process with quantization loss $\mathcal{L}_Q$. Block 3 visualizes the VQ process using multi-scale features, incorporating commitment loss $\mathcal{L}_C$ and our proposed Wasserstein loss $\mathcal{L}_W$.

In summary, even with a fixed feature distribution, methods like `Vanilla VQ` or those based on k-means and k-means++ fail to achieve distributional matching, leading to poor VQ performance. While `VQ+Linear` can achieve distributional matching with proper codebook initialization, its effectiveness heavily relies on this initialization. This limitation becomes potentially problematic when the feature distribution is unknown and changes dynamically during training. In contrast, our proposed *Wasserstein VQ* works independently of codebook initialization. Through explicit distributional matching regularization, it could maintain proper matching even as the feature distribution evolves dynamically. Notably, although the codebook distribution would be arbitrary during training, our quadratic Wasserstein distance—based on the Gaussian distribution assumption—effectively aligns the distributions and achieves the best VQ performance.

# 4 WASSERSTEIN VQ FOR VISUAL GENERATION

## 4.1 A PRELIMINARY: VQGAN

In this section, we examine the application of *Wasserstein VQ* within the framework of VQGAN (Esser et al., 2021). As illustrated in Block 1 of Figure 6, VQGAN combines several components: an encoder $E(\cdot)$, a decoder $D(\cdot)$, a quantizer $\mathcal{Q}(\cdot)$ with a learnable codebook $\{\mathbf{e}_k\}_{k=1}^{K}$, a VGG network $P(\cdot)$ (Simonyan & Zisserman, 2015), and a patch-based discriminator (Isola et al., 2017). As described earlier in Section 2.1, for an input image $\boldsymbol{x}$, the encoder processes the image to yield a spatial feature $\boldsymbol{z}_e = E(\boldsymbol{x}) \in \mathbb{R}^{h \times w \times d}$, where $(h, w)$ denotes the feature resolution. The quantizer converts $\boldsymbol{z}_e$ into a quantized feature $\boldsymbol{z}_q$, from which the decoder reconstructs the image as $\widehat{\boldsymbol{x}} = D(\boldsymbol{z}_q)$. To ensure high perceptual quality in the reconstructed images, the system employs both the VGG network and the patch-based discriminator (Esser et al., 2021; Johnson et al., 2016). The overall loss objective can be formulated as follows:

$$\mathcal{L} = \underbrace{\|\widehat{\boldsymbol{x}} - \boldsymbol{x}\|^2}_{\mathcal{L}_R} + \alpha_1 \underbrace{\|\text{sg}(\boldsymbol{z}_q) - \boldsymbol{z}_e\|}_{\mathcal{L}_Q} + \alpha_2 \underbrace{\|\text{sg}(\boldsymbol{z}_e) - \boldsymbol{z}_q\|}_{\mathcal{L}_C} + \underbrace{\|P(\widehat{\boldsymbol{x}}) - P(\boldsymbol{x})\|^2}_{\mathcal{L}_P} + \mathcal{L}_G(\boldsymbol{x}, \widehat{\boldsymbol{x}}), \quad (5)$$

where sg denotes the stop-gradient operation. $\mathcal{L}_R$, $\mathcal{L}_Q$, $\mathcal{L}_C$, $\mathcal{L}_P$, and $\mathcal{L}_G$ represent the reconstruction loss, quantization loss, commitment loss, VGG-based perceptual loss (Zhang et al., 2018), and GAN loss (Isola et al., 2017; Lim & Ye, 2017), respectively. $\alpha_1$ and $\alpha_2$ are hyper-parameters.

## 4.2 Multi-scale Vector Quantization

Drawing inspiration from VAR's coarse-to-fine token map design (Tian et al., 2024), we replace the vanilla VQ described in Section 2.1 with a multi-scale VQ approach. This modification is shown in Block 2 of Figure 6. The key difference is that multi-scale VQ employs a series of vanilla VQ steps, with each step processing feature vectors at increasingly higher resolutions.

To better understand the multi-scale VQ process, suppose we have a set of spatial features $\{z_i\}_{i=1}^T$ with a resolution of $(h, w)$ in Block 2, where $z_1$ is initialized with $z_e$. For creating coarse-to-fine image tokens, we extract multi-scale spatial features $\{g_i(z_i)\}_{i=1}^T$ using interpolation functions $g_i(\cdot)$ that reduces their resolutions to a set of smaller resolutions[6] $\{(h_i, w_i)\}_{i=1}^T$. These spatial features are processed by the vanilla VQ, yielding multi-scale image tokens $\{r_1, ..., r_T\}$ and multi-scale quantized features $\{\mathcal{Q}(g_i(z_i))\}_{i=1}^T$, as described in Block 3. The quantized features are then rescaled to their original resolutions via another set of interpolation functions $\widehat{g}_i(\cdot)$, denoted as $\widehat{z}_i = \widehat{g}_i(\mathcal{Q}(g_i(z_i)))$. Next, we take $z_{i+1} = z_i - \widehat{z}_i$ and perform residual quantization on $z_{i+1}$ (Lee et al., 2022).

The multi-scale VQ process operates sequentially, not in parallel. Through a multi-step VQ procedure, it derives the final quantized features $z_q = \sum_{i=1}^T \widehat{z}_i$, reducing the quantization error between $z_q$ and $z_e$. Notably, the commitment loss in multi-scale VQ differs from that in VQGAN (Esser et al., 2021) and can be written as:

$$\mathcal{L}_C = \sum_{i=1}^T \|g_i(z_i) - \mathcal{Q}(g_i(z_i))\|^2. \tag{6}$$

## 4.3 The Learning Objective

In this section, we focus on integrating our proposed *Wasserstein VQ* algorithm into the multi-scale VQ framework. We estimate the population mean and covariance of the multi-scale spatial features $\{g_i(z_i)\}_{i=1}^T$ and code vectors $\{\mathbf{e}_k\}_{k=1}^K$ by using their sample versions. To align the distributions between feature vectors and code vectors, we employ the quadratic Wasserstein distance $\mathcal{L}_W$ as defined in Equation 4. The overall objective function is:

$$\mathcal{L} = \mathcal{L}_R + \alpha_1 \mathcal{L}_Q + \alpha_2 \mathcal{L}_C + \alpha_3 \mathcal{L}_W + \mathcal{L}_P + \mathcal{L}_G, \tag{7}$$

where $\alpha_1$, $\alpha_2$ and $\alpha_3$ are hyperparameters.

## 5 Experiments

In this section, we empirically demonstrate the effectiveness of our proposed *Wasserstin VQ* algorithm in image reconstruction tasks. We conduct our experiments on the FFHQ (Karras et al., 2018) and ImageNet-1k (Deng et al., 2009) datasets. The PyTorch code will be made publicly available.

**Alternative Methods**  We evaluated our approach against several alternative methods: DQ-VAE (Huang et al., 2023a), DF-VQGAN (Ni et al., 2022), DiVAE (Shi et al., 2022), RQVAE (Lee et al., 2022), VQGAN (Esser et al., 2021), VQGAN-FC (Yu et al., 2022), VQGAN-EMA (Razavi et al., 2019), VQWAE (Vuong et al., 2023), MQVAE (Huang et al., 2023b), and VQGAN-LC (Zhu et al., 2024a). For detailed experimental settings, please refer to Appendix F.

**Evaluation Metrics**  Following prior works (Esser et al., 2021; Zhu et al., 2024a), we evaluate image reconstruction quality using multiple metrics: dataset-level Fréchet inception distance (rFID)(Heusel et al., 2017), feature-level learned perceptual image patch similarity (LPIPS)(Zhang et al., 2018), image-level peak signal-to-noise ratio (PSNR), and patch-level structural similarity index (SSIM).

**Main Results**  As shown in Tables 1 and 2, our proposed *Wasserstein VQ* outperforms all alternative methods on both datasets, demonstrating superior performance across all evaluation metrics at identical resolutions. Notably, *Wasserstein VQ* consistently maintains 100% codebook utilization, regardless of codebook size. This demonstrates that distributional matching effectively resolves the issue of codebook collapse.

---

[6]See the multi-scale resolution details in Appendix F.

Table 1: Reconstruction performance on the ImageNet-1K dataset. The term "Resolution" refers to the resolution of the spatial feature $z_e$, while "Utilization" represents codebook utilization $\mathcal{U}$. [†]: The codebook utilization is computed across the training dataset, [⋆]: The codebook utilization is computed across evaluation dataset. The symbol "-" indicates that no data point is provided.

| Method | Resolution | Codebook Size | Utilization (%) ↑ | rFID ↓ | LPIPS ↓ | PSNR ↑ | SSIM ↑ |
|---|---|---|---|---|---|---|---|
| DQVAE[†] | (16, 16) | 1,024 | - | 4.08 | - | - | - |
| DF-VQGAN[†] | (16, 16) | 12,288 | - | 5.16 | - | - | - |
| DiVAE[†] | (16, 16) | 16,384 | - | 4.07 | - | - | - |
| RQVAE[†] | (16, 16) | 16,384 | - | 3.20 | - | - | - |
| VQGAN[†] | (16, 16) | 16,384 | 3.4 | 5.96 | 0.17 | 23.3 | 52.4 |
| | (16, 16) | 50,000 | 1.1 | 5.44 | 0.17 | 22.5 | 52.5 |
| | (16, 16) | 100,000 | 0.5 | 5.44 | 0.17 | 22.3 | 52.5 |
| VQGAN-FC[†] | (16, 16) | 16,384 | 11.2 | 4.29 | 0.17 | 22.8 | 54.5 |
| | (16, 16) | 50,000 | 3.6 | 4.96 | 0.15 | 23.1 | 54.7 |
| | (16, 16) | 100,000 | 1.9 | 4.65 | 0.15 | 22.9 | 55.1 |
| VQGAN-EMA[†] | (16, 16) | 16,384 | 83.2 | 3.41 | 0.14 | 23.5 | 56.6 |
| | (16, 16) | 50,000 | 40.2 | 3.88 | 0.14 | 23.2 | 55.9 |
| | (16, 16) | 100,000 | 24.2 | 3.46 | 0.13 | 23.4 | 56.2 |
| VQGAN-LC[†] | (16, 16) | 16,384 | 99.9 | 3.01 | 0.13 | 23.2 | 56.4 |
| | (16, 16) | 50,000 | 99.9 | 2.75 | 0.13 | 23.8 | 58.4 |
| | (16, 16) | 100,000 | 99.9 | 2.62 | 0.12 | 23.8 | 58.9 |
| *Wasserstein VQ*[⋆] | (16, 16) | 16,384 | 100.0 | 2.28 | 0.12 | 24.43 | 63.5 |
| | (16, 16) | 50,000 | 100.0 | 2.07 | 0.12 | 24.67 | 64.4 |
| | (16, 16) | 100,000 | 100.0 | **1.94** | **0.11** | **24.76** | **64.8** |

Table 2: Reconstruction performance on the FFHQ dataset. The term "Resolution" refers to the resolution of the spatial feature $z_e$, while "Utilization" represents codebook utilization $\mathcal{U}$. [†]:The codebook utilization is computed across the training dataset, [⋆]: The codebook utilization is computed across evaluation dataset.

| Method | Resolution | Codebook Size | Utilization (%) ↑ | rFID ↓ | LPIPS ↓ | PSNR ↑ | SSIM ↑ |
|---|---|---|---|---|---|---|---|
| RQVAE[†] | (16, 16) | 2,048 | - | 7.04 | 0.13 | 22.9 | 67.0 |
| VQWAE[†] | (16, 16) | 1,024 | - | 4.20 | 0.12 | 22.5 | 66.5 |
| MQVAE[†] | (16, 16) | 1,024 | 78.2 | 4.55 | - | - | - |
| VQGAN[†] | (16, 16) | 16,384 | 2.3 | 5.25 | 0.12 | 24.4 | 63.3 |
| VQGAN-FC[†] | (16, 16) | 16,384 | 10.9 | 4.86 | 0.11 | 24.8 | 64.6 |
| VQGAN-EMA[†] | (16, 16) | 16,384 | 68.2 | 4.79 | 0.10 | 25.4 | 66.1 |
| VQGAN-LC[†] | (16, 16) | 100,000 | 99.5 | 3.81 | 0.08 | 26.1 | 69.4 |
| Wasserstein VQ[⋆] | (16, 16) | 16,384 | 100.0 | 3.52 | 0.08 | 27.07 | 74.4 |
| | (16, 16) | 50,000 | 100.0 | 3.35 | 0.08 | 27.26 | 74.9 |
| | (16, 16) | 100,000 | 100.0 | **3.18** | **0.07** | **27.32** | **74.9** |

**Ablation Studies** As shown in Table 3 in Appendix G, incorporating the Wasserstein distance $\mathcal{L}_W$ as an auxiliary loss function ($\alpha_3 = 0.3$) consistently outperforms the VQ algorithm without this term ($\alpha_3 = 0.0$). The improvement is evident in the reconstructed images' visual quality, particularly in the preservation of fine details, as demonstrated in Figure 10 in Appendix G.

# 6 CONCLUSION

This paper examines vector quantization (VQ) from a distributional perspective. We introduce three key evaluation criteria and demonstrate empirically that optimal VQ results emerge when the distributions of continuous feature vectors and code vectors are identical. Our theoretical analysis confirms this finding, emphasizing the crucial role of distributional alignment in effective VQ. Based on these insights, we propose using the quadratic Wasserstein distance to achieve alignment, leveraging its computational efficiency under a Gaussian hypothesis. This approach achieves near-full codebook utilization while significantly reducing quantization error. Our method successfully addresses both training instability and codebook collapse, leading to improved downstream image reconstruction performance. However, due to limited GPU resources, we were unable to conduct image generation experiments.

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

# Appendix

## Table of Contents

## A   OPTIMAL SUPPORT OF THE CODEBOOK DISTRIBUTION

*Proof of Theorem 1.* First, we assume $\overline{\mathrm{supp}(\mathcal{P}_B)} = \overline{\mathrm{supp}(\mathcal{P}_A)}$. Then for any $\boldsymbol{z} \in \mathrm{supp}(\mathcal{P}_A)$, there exist a sequence of points in $\mathrm{supp}(\mathcal{P}_B)$ that converge to $\boldsymbol{z}$. Let $\{\boldsymbol{e}_k\}_{k=1}^K$ be $K$ code vectors independently generated from $\mathcal{P}_B$. Then the empirical distribution of $\{\boldsymbol{e}_k\}_{k=1}^K$ tends to $\mathcal{P}_B$ as the size $K$ tends to infinity. Since $\Omega = \mathrm{supp}(\mathcal{P}_A)$ is a bounded region, we have the following:

$$\sup_{\boldsymbol{z} \in \overline{\mathrm{supp}(\mathcal{P}_A)}} \min_k \|\boldsymbol{z} - \boldsymbol{e}_k\|^2 = \sup_{\boldsymbol{z} \in \overline{\mathrm{supp}(\mathcal{P}_B)}} \min_k \|\boldsymbol{z} - \boldsymbol{e}_k\|^2 \xrightarrow{p} 0, \quad \text{as } K \to \infty.$$

This quantity is an upper bound on the quantization error $\mathcal{E}(\{\boldsymbol{z}_i\}; \{\boldsymbol{e}_k\})$. Thus,

$$\sup_{\{\boldsymbol{z}_i\} \subseteq \Omega} \mathcal{E}\left(\{\boldsymbol{z}_i\}_{i=1}^N; \{\boldsymbol{e}_k\}_{k=1}^K\right) \leq \sup_{\boldsymbol{z} \in \overline{\Omega}} \min_k \|\boldsymbol{z} - \boldsymbol{e}_k\|^2 \xrightarrow{p} 0, \quad \text{as } K \to \infty.$$

This demonstrates that $\mathcal{P}_B$ has vanishing quantization error asymptotically. Furthermore, for any $K$ code vectors $\{\boldsymbol{e}_k\}_{k=1}^K$ independently drawn from $\mathcal{P}_B$, we have $\{\boldsymbol{e}_k\}_{k=1}^K \subseteq \overline{\Omega}$. Since the empirical distribution of $\{\boldsymbol{z}_i\}_{i=1}^N$ tends to $\mathcal{P}_A$ as the feature sample size $N$ tends to infinity, we can easily show that for any fixed $\{\boldsymbol{e}_k\}_{k=1}^K \subseteq \overline{\Omega}$, the codebook utility rate satisfies

$$\mathcal{U}\left(\{\boldsymbol{z}_i\}_{i=1}^N, \{\boldsymbol{e}_k\}_{k=1}^K\right) \xrightarrow{p} 1, \quad \text{as } N \to \infty.$$

This shows that $\{\boldsymbol{e}_k\}_{k=1}^K$ attains full utilization asymptotically, and thus $\mathcal{P}_B$ attains full utilization asymptotically.

On the other hand, we assume $\mathcal{P}_B$ attains full utilization and vanishing quantization error asymptotically. Then we first claim that $\overline{\mathrm{supp}(\mathcal{P}_A)} \subseteq \overline{\mathrm{supp}(\mathcal{P}_B)}$. Since $\mathcal{P}_B$ has vanishing quantization error asymptotically, then for any $\boldsymbol{z} \in \mathrm{supp}(\mathcal{P}_A)$, there exist a sequence of points in $\mathrm{supp}(\mathcal{P}_B)$ that converge to $\boldsymbol{z}$. This implies that $\mathrm{supp}(\mathcal{P}_A) \subseteq \overline{\mathrm{supp}(\mathcal{P}_B)}$ and thus $\overline{\mathrm{supp}(\mathcal{P}_A)} \subseteq \overline{\mathrm{supp}(\mathcal{P}_B)}$.

To show $\overline{\text{supp}(\mathcal{P}_B)} = \overline{\text{supp}(\mathcal{P}_A)}$, it remains to show $\text{supp}(\mathcal{P}_B) \subseteq \overline{\text{supp}(\mathcal{P}_A)}$. In fact, if $\text{supp}(\mathcal{P}_B) \subseteq \overline{\text{supp}(\mathcal{P}_A)}$ does not hold, then there exists an open region $\mathcal{R} \subseteq \text{supp}(\mathcal{P}_B) - \overline{\text{supp}(\mathcal{P}_A)}$ such that $\mathcal{P}_B(\mathcal{R}) > 0$ and

$$\min_{\boldsymbol{z} \in \text{supp}(\mathcal{P}_A), \boldsymbol{z}' \in \mathcal{R}} \|\boldsymbol{z} - \boldsymbol{z}'\| \geq \epsilon_0$$

for some $\epsilon_0 > 0$. Since $\text{supp}(\mathcal{P}_A) \subseteq \overline{\text{supp}(\mathcal{P}_B)}$, then there exists a sufficiently large $K_0$ such that the event

$$\left\{ \text{Generating} \{\boldsymbol{e}_k\}_{k=1}^{K_0} \text{ i.i.d. from } \mathcal{P}_B \text{ s.t. } \{\boldsymbol{e}_k\} \subseteq \text{supp}(\mathcal{P}_A), \sup_{\boldsymbol{z} \in \text{supp}(\mathcal{P}_A)} \min_k \|\boldsymbol{z} - \boldsymbol{e}_k\| < \epsilon_0 \right\} \quad (8)$$

has some positive probability $C > 0$. Then with a positive probability of at least $C \cdot \mathcal{P}_B(\mathcal{R})$, we can pick the first $K_0$ code vectors from Equation (8) and the $(K_0 + 1)$th code vector from $\mathcal{R}$. For any such codebook of size $K_0 + 1$, we know the $(K_0 + 1)$th code vector will never be used regardless of the choice of the feature set $\{\boldsymbol{z}_i\}$. Therefore, the codebook utilization

$$\sup_{\{\boldsymbol{z}_i\}} \mathcal{U}\left( \{\boldsymbol{e}_k\}_{k=1}^{K_0+1}; \{\boldsymbol{z}_i\} \right) \leq \frac{K_0}{K_0 + 1} < 1.$$

This contradicts the property that $\mathcal{P}_B$ attains full utilization asymptotically. Thus, $\text{supp}(\mathcal{P}_B) \subseteq \overline{\text{supp}(\mathcal{P}_A)}$ must hold. This concludes the proof. $\qquad\square$

## B  STATISTICAL DISTANCES OVER GAUSSIAN DISTRIBUTIONS

We first introduce the definition of Wasserstein distance.

**Definition 4.** *The Wasserstein distance or earth-mover distance with p norm is defined as below:*

$$W_p(\mathbb{P}_r, \mathbb{P}_g) = \big( \inf_{\gamma \in \Pi(\mathbb{P}_r, \mathbb{P}_g)} \mathbb{E}_{(x,y)\sim\gamma} \big[ \|x - y\|^p \big] \big)^{1/p} . \quad (9)$$

where $\Pi(\mathcal{P}_r, \mathcal{P}_g)$ denotes the set of all joint distributions $\gamma(x, y)$ whose marginals are $\mathcal{P}_r$ and $\mathcal{P}_g$ respectively. Intuitively, when viewing each distribution as a unit amount of earth/soil, the Wasserstein distance (also known as earth-mover distance) represents the minimum cost of transporting "mass" from $x$ to $y$ to transform distribution $\mathcal{P}_r$ into distribution $\mathcal{P}_g$. When $p = 2$, this is called the quadratic Wasserstein distance.

In this paper, we achieve distributional matching using the quadratic Wasserstein distance under Gaussian distribution assumptions. We also examine other statistical distribution distances as potential loss functions for distributional matching and compare them with the Wasserstein distance. Specifically, we provide the Kullback-Leibler divergence and the Bhattacharyya distance over Gaussian distributions in Lemma 5 and Lemma 6. Both distances require full-rank covariance matrices, which makes them unsuitable for distributional matching in practical applications. In contrast, our quadratic Wasserstein distance-based loss function does not have this limitation.

**Lemma 5** (Kullback-Leibler divergence (Lindley & Kullback, 1959)). *Suppose two random variables* $\mathbf{Z}_1 \sim \mathcal{N}(\boldsymbol{\mu}_1, \boldsymbol{\Sigma}_1)$ *and* $\mathbf{Z}_2 \sim \mathcal{N}(\boldsymbol{\mu}_2, \boldsymbol{\Sigma}_2)$ *obey multivariate normal distributions, then Kullback-Leibler divergence between* $\mathbf{Z}1$ *and* $\mathbf{Z}_2$ *is:*

$$D_{\text{KL}}(\mathbf{Z}_1, \mathbf{Z}_2) = \frac{1}{2}((\boldsymbol{\mu}_1 - \boldsymbol{\mu}_2)^T \boldsymbol{\Sigma}_2^{-1}(\boldsymbol{\mu}_1 - \boldsymbol{\mu}_2) + \text{tr}(\boldsymbol{\Sigma}_2^{-1}\boldsymbol{\Sigma}_1 - \mathbf{I}) + \ln \frac{\det \boldsymbol{\Sigma}_2}{\det \boldsymbol{\Sigma}_1}).$$

**Lemma 6** (Bhattacharyya Distance (Bhattacharyya, 1943)). *Suppose two random variables* $\mathbf{Z}_1 \sim \mathcal{N}(\boldsymbol{\mu}_1, \boldsymbol{\Sigma}_1)$ *and* $\mathbf{Z}_2 \sim \mathcal{N}(\boldsymbol{\mu}_2, \boldsymbol{\Sigma}_2)$ *obey multivariate normal distributions,* $\boldsymbol{\Sigma} = \frac{1}{2}(\boldsymbol{\Sigma}_1 + \boldsymbol{\Sigma}_2)$, *then bhattacharyya distance between* $\mathbf{Z}1$ *and* $\mathbf{Z}_2$ *is:*

$$\mathcal{D}_B(\mathbf{Z}_1, \mathbf{Z}_2) = \frac{1}{8}(\boldsymbol{\mu}_1 - \boldsymbol{\mu}_2)^T \boldsymbol{\Sigma}^{-1}(\boldsymbol{\mu}_1 - \boldsymbol{\mu}_2) + \frac{1}{2} \ln \frac{\det \boldsymbol{\Sigma}}{\sqrt{\det \boldsymbol{\Sigma}_1 \det \boldsymbol{\Sigma}_2}}.$$

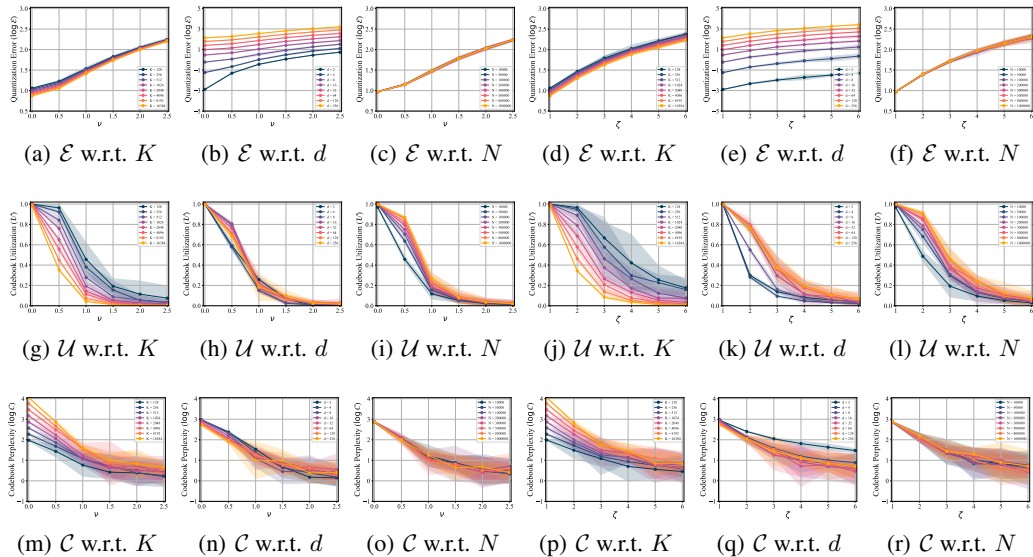

(a) $\mathcal{E}$ w.r.t. $K$    (b) $\mathcal{E}$ w.r.t. $d$    (c) $\mathcal{E}$ w.r.t. $N$    (d) $\mathcal{E}$ w.r.t. $K$    (e) $\mathcal{E}$ w.r.t. $d$    (f) $\mathcal{E}$ w.r.t. $N$

(g) $\mathcal{U}$ w.r.t. $K$    (h) $\mathcal{U}$ w.r.t. $d$    (i) $\mathcal{U}$ w.r.t. $N$    (j) $\mathcal{U}$ w.r.t. $K$    (k) $\mathcal{U}$ w.r.t. $d$    (l) $\mathcal{U}$ w.r.t. $N$

(m) $\mathcal{C}$ w.r.t. $K$    (n) $\mathcal{C}$ w.r.t. $d$    (o) $\mathcal{C}$ w.r.t. $N$    (p) $\mathcal{C}$ w.r.t. $K$    (q) $\mathcal{C}$ w.r.t. $d$    (r) $\mathcal{C}$ w.r.t. $N$

Figure 7: Quantitative analyses of the criterion triple when $\mathcal{P}_A$ and $\mathcal{P}_B$ are uniform distributions.

## C QUANTITATIVE ANALYSES WHEN CODEBOOK DISTRIBUTION AND FEATURE DISTRIBUTION ARE UNIFROM DISTRIBUTIONS

As discussed in Section 2.3, we conclude that the optimal criterion triple is achieved when $\mathcal{P}_A$ and $\mathcal{P}_B$ exhibit identical distributions. This conclusion holds when $\mathcal{P}_A$ and $\mathcal{P}_B$ are derived from other distributions, such as the uniform distribution. As shown in Figure 7, we sample a set of feature vectors $\{z_i\}_{i=1}^N$ from the distribution $\text{Unif}(-1, 1)$ and a set of code vectors $\{e_k\}_{k=1}^K$ from $\text{Unif}(\nu - 1, \nu + 1)$, where $\nu$ is selected from the set $\{0.0, 0.5, 1.0, 1.5, 2.0, 2.5\}$ or from $\text{Unif}(-\zeta, \zeta)$, with $\zeta$ drawn from the set $\{1, 2, 3, 4, 5, 6\}$. We observe that when $\mu = 0$ or $\zeta = 1$—indicating that $\mathcal{P}_A$ and $\mathcal{P}_B$ have identical distributions—the performance in terms of the criterion triple is optimal, achieving the lowerest $\mathcal{E}$, the highest $\mathcal{U}$, and the largest $\mathcal{C}$ across all tested values of $K, d, N$. Therefore, we conclude that our quantitative analyses are distribution-agnostic and can be generalized to other distributions.

## D QUANTIZATION ERROR ANALYSES UNDER THE DISTRIBUTION MATCHING

As discussed in Section 2.3 and 2.4, the minimum $\mathcal{E}$ occurs when the distributions $\mathcal{P}_A$ and $\mathcal{P}_B$ are identical. In this section, we explore other factors influencing $\mathcal{E}$ as part of the supplementary analyses in Section 2.3. We first consider both $\mathcal{P}_A$ and $\mathcal{P}_B$ to be Gaussian distributions. As illustrated in Figure 8, we sample a set of feature vectors $\{z_i\}_{i=1}^N$ along with a set of code vectors $\{e_k\}_{k=1}^K$ from the distribution $\mathcal{N}(\mu * \mathbf{1}_d, I_d)$, where $\mu$ is selected from the set $\{0.0, 0.5, 1.0, 1.5, 2.0, 2.5\}$, or from the distribution $\mathcal{N}(\mathbf{0}_d, \sigma^2 I_d)$, where $\sigma$ is drawn from the set $\{1, 2, 3, 4, 5, 6\}$. From the Figure 8(a) to 8(f), we observe that $\mathcal{E}$ remains constant as $\mu$ increases, while it increases as $\sigma$ increases. Additionally, we find that the feature size $N$ does not significantly impact $\mathcal{E}$. A larger codebook size $K$ results in a slight decrease in $\mathcal{E}$, whereas an increase in the codebook dimension $d$ leads to a markedly larger $\mathcal{E}$. Consequently, the minimum value of $\mathcal{E}$ is influenced by the Gaussian covariance matrix $\sigma^2 I$, the codebook size $K$, and the feature dimension $d$. This analysis underscores the nuanced interplay of these parameters in determining the optimal performance of our model under Gaussian distributional assumptions.

We can arrive at a nearly identical conclusion when $\mathcal{P}_A$ and $\mathcal{P}_B$ are derived from other distributions, such as the uniform distribution. As shown in Figure 9, we sample a set of feature vectors $\{z_i\}_{i=1}^N$ along with a set of code vectors $\{e_k\}_{k=1}^K$ from the distribution $\text{Unif}(\nu - 1, \nu + 1)$, where $\nu$ is selected from the set $\{0.0, 0.5, 1.0, 1.5, 2.0, 2.5\}$, or from the distribution $\text{Unif}(-\zeta, \zeta)$, with $\zeta$ drawn from the set $\{1, 2, 3, 4, 5, 6\}$. Similarly, the minimum value of $\mathcal{E}$ is influenced by the $\zeta$, the codebook size $K$, and the feature dimension $d$.

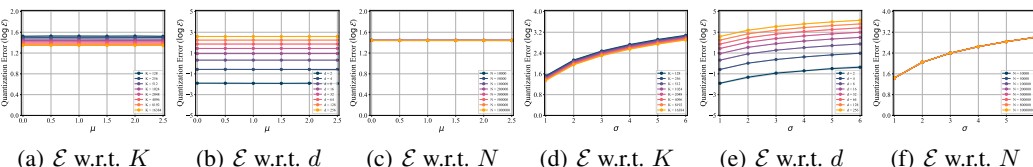

(a) $\mathcal{E}$ w.r.t. $K$    (b) $\mathcal{E}$ w.r.t. $d$    (c) $\mathcal{E}$ w.r.t. $N$    (d) $\mathcal{E}$ w.r.t. $K$    (e) $\mathcal{E}$ w.r.t. $d$    (f) $\mathcal{E}$ w.r.t. $N$

Figure 8: Visualization of quantization Error when $\mathcal{P}_A$ and $\mathcal{P}_B$ are Gaussian distributions.

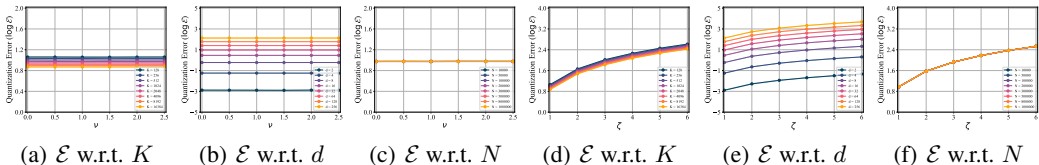

(a) $\mathcal{E}$ w.r.t. $K$    (b) $\mathcal{E}$ w.r.t. $d$    (c) $\mathcal{E}$ w.r.t. $N$    (d) $\mathcal{E}$ w.r.t. $K$    (e) $\mathcal{E}$ w.r.t. $d$    (f) $\mathcal{E}$ w.r.t. $N$

Figure 9: Visualization of quantization Error when $\mathcal{P}_A$ and $\mathcal{P}_B$ are Uniform distributions.

## E  THE DETAILS OF SYNTHETIC EXPERIMENTS

### E.1  EXPERIMENTAL DETAILS IN SECTION 2.3 AND APPENDIX C

**Qualitative Analyses**  As depicted in Figure 3 in Section 2.3, we conduct a qualitative analyses of the criterion triple. Specifically, we sample a set of feature vectors $\{z_i\}_{i=1}^N$ from within the red circle, and a collection of code vectors $\{e_k\}_{k=1}^K$ from within the green circle, with parameters set to $K = 400$, $N = 10000$ and $d = 2$ for the calculation of the criterion triple $(\mathcal{E}, \mathcal{U}, \mathcal{C})$. For the visualization, we select 10% of the feature vectors and 90% of the code vectors for plotting.

**Quantitative Analyses**  As illustrate in Figure 4 in Section 2.3, we undertake comprehensive quantitative analyses centered around the criterion triple $(\mathcal{E}, \mathcal{U}, \mathcal{C})$. In these analyses, we assume that $\mathcal{P}_A$ and $\mathcal{P}_B$ are Gaussian distributions, from which we sample a set of feature vectors $\{z_i\}_{i=1}^N$ and a collection of code vectors $\{e_k\}_{k=1}^K$. The default parameters are set to $N = 200,000$, $K = 1024$, and $d = 32$ for all figures unless otherwise specified. For instance, in Figure 4(a), $N$ and $d$ are taken at their default values, while the $K$ is varied within the set $\{128, 256, 512, 1024, 2048, 4096, 8192, 16284\}$. Additionally, each synthetic experiment is repeated five times, and the average results are reported, along with the calculation of 95% confidence intervals. In all figures, mean results are represented by points, while the confidence intervals are shown as shaded areas. Identical parameter settings are employed when $\mathcal{P}_A$ and $\mathcal{P}_B$ are uniform distributions, as illustrated in Figure 7 in Appendix C.

### E.2  EXPERIMENTAL DETAILS IN SECTION 3.2

We provide experimental details of Figure 5 in Section 3.2. In our experimental setup, we evaluate five distinct VQ algorithms using the criterion triple $(\mathcal{E}, \mathcal{U}, \mathcal{C})$. All experiments run on a single NVIDIA A100 GPU, with a codebook size $K$ of 16,384 and dimensionality $d$ of 16 across all algorithms. Each algorithm trains for 2,000 steps, with 20,000 feature vectors sampled from the specified Gaussian distribution at each step. For *Wasserstein VQ*, Vanilla VQ, and VQ + MLP, we use the SGD optimizer for training. For VQ EMA and Online Clustering, we use classical clustering algorithms—$k$-means (Bradley & Fayyad, 1998) and $k$-means++(Arthur & Vassilvitskii, 2007)—to update code vectors.

### E.3  EXPERIMENTAL DETAILS IN APPENDIX D

For the Figure 8 and 9 in Appendix D, all figures utilize the default parameters as specified in the quantitative analyses presented in Appendix C when calculating the criterion triple $(\mathcal{E}, \mathcal{U}, \mathcal{C})$.

## F  IMAGE RECONSTRUCTION EXPERIMENTAL DETAILS

In the image reconstruction task, our proposed *Wasserstein VQ* adopts the same encoder and decoder as the original VQGAN (Esser et al., 2021). Input images are processed at a resolution of $(256, 256)$. The encoder, a U-Net (Ronneberger et al., 2015), downscales the input image by a factor of 16, yielding a spatial feature $z_e$ with the resolution of $(16, 16)$. This spatial feature is quantized while maintaining the same resolution, then fed into the decoder (also a U-Net) for image reconstruction. To generate coarse-to-fine image tokens, we extract multi-scale spatial features using an interpolation function that reduces their resolutions to progressively smaller sizes. We follow the token map design from VAR (Tian et al., 2024), setting $T$ to 10, with multi-scale spatial feature resolutions of $(1, 1)$, $(2, 2)$, $(3, 3)$, $(4, 4)$, $(5, 5)$, $(6, 6)$, $(8, 8)$, $(10, 10)$, $(13, 13)$, and $(16, 16)$. For all experiments, we set $\alpha_1 = \alpha_2 = 0.2$ and $\alpha_3 = 0.3$. We employ the Adam optimizer (Kingma & Ba, 2014) with an initial learning rate of $5e^{-4}$, applying a half-cycle cosine decay after a 5-epoch linear warm-up phase. We train for 20 epochs on ImageNet-1k using 8 Nvidia H20 GPUs, and for 200 epochs on the FFHQ dataset using 4 Nvidia H20 GPUs.

## G  ABLATION RECONSTRUCTION RESULTS

To evaluate the effectiveness of our proposed Wasserstein distance $\mathcal{L}_W$, we conducted ablation studies. As shown in Table 3, using the Wasserstein distance $\mathcal{L}_W$ as an auxiliary loss function ($\alpha_3 = 0.3$) consistently outperforms the VQ algorithm without it ($\alpha_3 = 0.0$). This improvement also can be observed in the reconstructed images' visual quality, especially in the preservation of fine details, as shown in Figure 10.

Table 3: Ablation studies of the Wasserstein distance on the ImageNet-1K dataset. The term "Utilization" refers to codebook utilization $\mathcal{U}$, calculated across the evaluation dataset. $\alpha_3 = 0.3$ indicates the incorporation of the Wasserstein distance in the VQ algorithm, while $\alpha_3 = 0.0$ signifies the exclusion of the Wasserstein distance. $\uparrow$ indicates improvements.

| Method | Codebook Size | Utilization (%) $\uparrow$ | rFID $\downarrow$ | LPIPS $\downarrow$ | PSNR $\uparrow$ | SSIM $\uparrow$ |
|---|---|---|---|---|---|---|
| $\alpha_3 = 0.0$ | 16,384 | 89.3 | 2.74 | 0.13 | 23.97 | 61.6 |
| $\alpha_3 = 0.3$ | 16,384 | 100.0 $\uparrow_{10.7}$ | 2.28 $\uparrow_{0.46}$ | 0.12 $\uparrow_{0.01}$ | 24.43 $\uparrow_{0.46}$ | 63.5 $\uparrow_{1.9}$ |
| $\alpha_3 = 0.0$ | 50,000 | 73.5 | 2.48 | 0.13 | 24.23 | 62.6 |
| $\alpha_3 = 0.3$ | 50,000 | 100.0 $\uparrow_{26.5}$ | 2.07 $\uparrow_{0.41}$ | 0.12 $\uparrow_{0.01}$ | 24.67 $\uparrow_{0.44}$ | 64.4 $\uparrow_{1.8}$ |
| $\alpha_3 = 0.0$ | 100,000 | 61.8 | 2.27 | 0.12 | 24.30 | 62.8 |
| $\alpha_3 = 0.3$ | 100,000 | 100.0 $\uparrow_{38.2}$ | **1.94** $\uparrow_{0.33}$ | **0.11** $\uparrow_{0.01}$ | **24.76** $\uparrow_{0.46}$ | **64.8** $\uparrow_{2.0}$ |

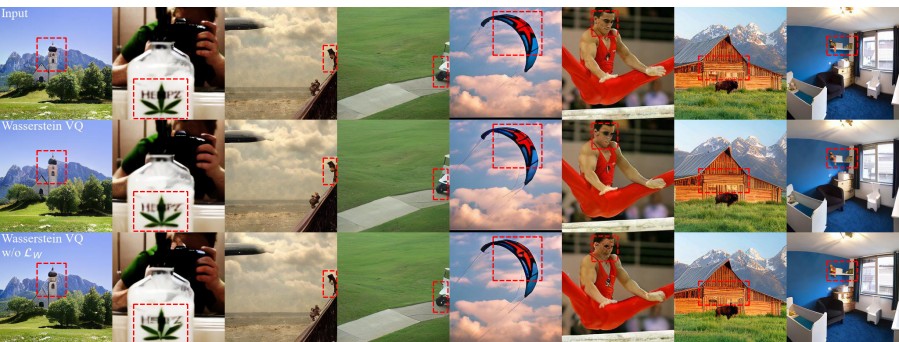

Figure 10: Visualization of Reconstructed Images. The first row exhibits the original input images at a resolution of $256 \times 256$ pixels, the second row shows the reconstruction results from the *Wasserstein VQ* method, while the third row presents the outcomes without the incorporation of the Wasserstein distance.

## H  ADDITIONAL EXPLANATIONS

### H.1  EXPLANATIONS ON THE CRITERION 2 AND 3

This section offers visual elucidations for Criterion 2 and 3 that are defined in Section 2.2. As depicted in the Figure 11(a) and 11(b), the values of $\mathcal{U}$ are 50% and 100%, respectively. This discrepancy arises because, in Figure 11(a) only half of code vectors' utilization $p_k$ exceeds zero

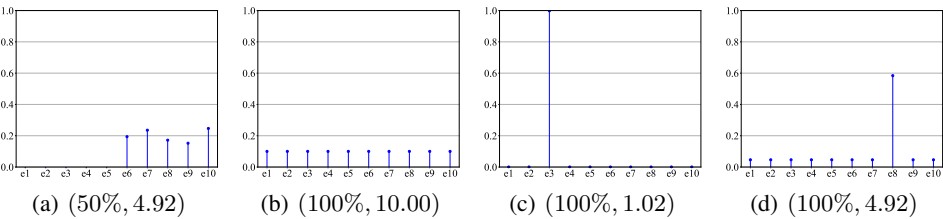

$$\text{(a) } (50\%, 4.92) \qquad \text{(b) } (100\%, 10.00) \qquad \text{(c) } (100\%, 1.02) \qquad \text{(d) } (100\%, 4.92)$$

Figure 11: Visualization of the evaluation criteria $(\mathcal{U}, \mathcal{C})$.

(as stipulated by in Criterion 3), whereas in Figure 11(b), the utilization $p_k$ of of all code vectors surpasses zero[7]. It is clear that $\mathcal{U}$ quantifies the *completeness* of codebook utilization. However, $\mathcal{U}$ remains insufficient to evaluate the degree of codebook collapse, as it fails to address the scenario depicted in Figure 11(c). Although all code vectors are utilized, the code vector $e_3$ excessively dominates the codebook utilization, resulting in an extreme imbalance. This imbalanced codebook utilization can be considered a form of codebook collapse, thereby not aligning with our desired outcome. This observation motivates the proposal of Criterion 3, which is capable of gauging the *imbalance* or *uniformity* inherent in codebook utilization.

When compared in Figure 11(b) and 11(c), the value of $\mathcal{C}$ are 10.00 and 1.02, respectively, demonstrating that Criterion 3 is capable of distinguishing the imbalance of code vector utilization $p_k$ under conditions where cases share the same $\mathcal{U}$. Additionally, Criterion 3 categorizes Figure 11(c) as indicative of codebook collapse, as the value $\mathcal{C}$ nearly reaches its minimum of 1.0, a result that resonates with our intuitive interpretation. However, it is essential to note that Criterion 3 alone does not suffice to evaluate the degree of codebook collapse. When scrutinizing Figure 11(a) and 11(d), despite the identical $\mathcal{C}$, there exists a stark disparity in $\mathcal{U}$. This observation underscores that the value of $\mathcal{C}$ is inadequate for quantifying the proportion of actively utilized code vectors.

In this paper, we adopt the combination of Criterion 2 and 3 to quantitatively assess the extent of codebook collapse. A robust mitigation of codebook collapse is indicated solely when both $\mathcal{U}$ and $\mathcal{C}$ exhibit substantial values.

## H.2 Explanations on the Prototypical Study in Section 2.3

This section interprets the experimental findings presented in Figure 3. The VQ process relies on nearest neighbor search for code vector selection. As evident from Figure 3(a) to 3(d), actively selected code vectors are predominantly those located in close proximity to or within the feature distribution, while distant ones remain unselected. This leads to highly uneven code vector utilization $p_k$, with those closer to the feature distribution being excessively used. This elucidates the significantly low $\mathcal{U}$ and $\mathcal{C}$ observed in Figure 3(a). Furthermore, a notable quantization error, e.g., $\mathcal{E} = 1.19$ in Figure 3(a), arises when the codebook and feature distributions are mismatched, forcing feature vectors outside the codebook to settle for distant code vectors. Conversely, as the disk centers align, leading to a closer match between the two distributions, an increased number of code vectors become actively engaged. Additionally, code vectors are utilized more uniformly, and feature vectors can select nearer counterparts. This accounts for the improvement of criterion triple values towards optimality as the distributions align.

Analogously, we can employ nearest neighbor search to interpret the second case. When code vectors are distributed within the range of feature vectors, as illustrated in Figure 3(e) and Figure 3(f), the majority of code vectors would be actively utilized, ensuring high $\mathcal{U}$. However, the utilization of these code vectors is not uniform; code vectors on the periphery of the codebook distribution are more frequently used, leading to relatively low $\mathcal{C}$. Feature vectors on the periphery will have larger distances to their nearest code vectors, resulting in higher $\mathcal{E}$. Conversely, when feature vectors fall within the range of code vectors, as depicted in Figure 3(g) and Figure 3(h), outer code vectors remain largely unused, leading to a lower $\mathcal{U}$ and $\mathcal{C}$. Since only inner code vectors are active, each feature vector can find a nearby counterpart, maintaining low $\mathcal{E}$.

## H.3 Why k-means-Based VQ Methods Fail to Achieve Distributional Matching

---

[7]the concept of $p_k$ is analogous to that of sub-word frequency over the text corpus in the natural language processing field.

In this section, we offer visual illustrations to elucidate why $k$-means-based VQ methods fall short in achieving distributional matching. $k$-means-based VQ algorithm was originally proposed (van den Oord et al., 2017) and subsequently employed (Razavi et al., 2019). However, a widely acknowledged limitation of this approach is the significant issue of codebook collapse, particularly when the codebook size, $K$, is large (Dhariwal et al., 2020; Takida et al., 2022; Yu et al., 2022; Lee et al., 2022).

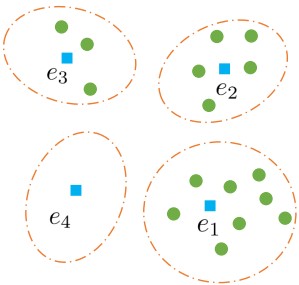

Figure 12: Visualization of $k$-mean assignment step. A green circle and a blue square represent the feature and code vectors, respectively.

To provide a deeper understanding of this intrinsic issue, we visually depict the $k$-means assignment step for VQ in Figure 12. The $k$-means algorithm initially partitions the feature space into Voronoi cells by assigning each feature vector to the nearest code vector based on Euclidean distance. In this step, nine feature vectors are assigned to $e_1$, whereas no feature vector is assigned to $e_4$. Subsequently, these code vectors, acting as the clustering centers, are updated using an exponential moving average of the assigned feature vectors. This update mechanism presents a critical challenge: since $e_4$ is never selected, it remains unupdated. In practical applications, especially when the codebook size $K$ is substantial, a majority of code vectors remain unutilized and unupdated. This phenomenon highlights the inherent difficulty of $k$-means-based VQ methods in learning an effective and representative codebook for distributional matching.

In addition to the explanations provided, we also present empirical evidence demonstrating that $k$-means-based VQ methods fail to achieve distributional matching. For detailed insights, please refer to Section 3.2.

