# OpenReview forum: "Vector Quantization By Distribution Matching"
_ICLR.cc/2025/Conference — Submitted to ICLR 2025_

### Official Review · Reviewer_YHdN · 2024-10-28

**Soundness:** 2
**Presentation:** 2
**Contribution:** 2
**Rating:** 5
**Confidence:** 4

**Summary:**

In the paper, the authors highlighted that a low utilization rate of the codebook will impair the training stability  of Vector Quantization (VQ), and subsequently proposed to enhance this utilization rate by aligning the distributions of feature vectors with those of the codebook.

**Strengths:**

1) To enhance the utilization rate of codebook, this paper proposed to align the distributions of feature vectors with those of the codebook.

2) Gaussian distributions are adopted  to model the feature and code vectors.

 3) Wasserstein  distance is used d to measure the distance between distributions.

**Weaknesses:**

1) Using Gaussian distributions to model the feature and code vectors is not reasonable, as these vectors typically belong to diverse classes with more intricate distributions. From a statistical perspective, modeling the data with Gaussian mixtures is more appropriate. Consequently, it is reasonable to raise doubts regarding the effectiveness of the proposed distribution matching method.

2) Improving the utilization rate of  code vectors in a  codebook (or called dictionary) is not a novel problem, which has been early studied in dictionary learning.

**Questions:**

1) Is the low utilization rate of code vectors a result of the gradient backpropagation problem (as argued by the authors in line 049), or is it due to the insufficient diversity of training samples relative to the large size of the codebook? I lean towards the latter explanation, and therefore consider the low utilization rate is *not* a pivotal issue that significantly impacts the final performance of vector quantization.

2) In Lines 062-068, the authors should  delve into the reasons behind the low code vector utilization rate observed in existing methods. For instance, in my view, the codebook initialization with k-means (Zhu et al. 2024) should work well in distribution matching, but performed worse in the paper. Could this be attributed to the issue of codebook utilization rate, or other factors, such as differences in network structures?

3) A codebook with a high utilization rate should closely match the distribution of feature vectors. Therefore, is it necessary to incorporate an additional distribution matching approach based on the Wasserstein distance?

4) As previously mentioned, modeling the entire set of feature vectors using a single Gaussian distribution is not reasonable. Instead, Gaussian mixtures are recommended as a more suitable alternative.

---

> ### Author Response · Authors · 2024-11-26
> **Response to reviewer YHdN (1/3)**
>
> **R 4.1** In the paper, the authors highlighted that a low utilization rate of the codebook will impair the training stability of Vector Quantization (VQ), and subsequently proposed to enhance this utilization rate by aligning the distributions of feature vectors with those of the codebook.
>
> Thank you for summarizing our paper. However, it is important to clarify that we have never claimed that **a low utilization rate of the codebook impairs the training stability of VQ.** Our paper examines two distinct issues in VQ: codebook collapse and training stability, without suggesting any causal relationship between them. Rather, we elucidate that training stability arises from the gradient discrepancy between $z_q$ and $z_e$, and we address this issue by achieving minimal quantization error, thereby establishing a stable VQ system with minimal information loss.
>
> **R 4.2** Using Gaussian distributions to model the feature and code vectors is not reasonable, as these vectors typically belong to diverse classes with more intricate distributions. From a statistical perspective, modeling the data with Gaussian mixtures is more appropriate. Consequently, it is reasonable to raise doubts regarding the effectiveness of the proposed distribution matching method.
>
> We acknowledge that relying on the assumption of a Gaussian distribution may have undesired effects. However, it is worth noting that this assumption is commonly employed in various practices. For instance, the Fréchet inception distance (FID), a widely used metric for evaluating generated images from generative models like diffusion models, assumes that both the generated images and the ground-truth images follow a Gaussian distribution. Despite its limitations, FID remains a popular choice for measuring the distance between images in different domains.
>
> The real distribution of the feature and code vectors would be exceedingly intricate and often defies precise mathematical formulation. Our selection of the Gaussian assumption is primarily motivated by the Gaussian distribution’s inherent simplicity. Concurrently, the central limit theorem plays a role in ensuring that learned feature vectors and code vectors follow a Gaussian distribution provided a large sample size and a large codebook size. Indeed, the Gaussian assumption has been used in many studies such as in GANs, where the latent distribution is assumed to be Gaussian.
>
> The effectiveness of the Gaussian assumption is also evident in our experiments. For instance, in Section 3.2, despite initializing the codebook with a standard Gaussian distribution, the optimization process of the codebook distribution would inherently be arbitrary during training. Nevertheless, our quadratic Wasserstein distance metric—which is based on the Gaussian distribution assumption—still effectively aligns the distributions and achieves the best VQ performance.
>
> **R 4.3** Improving the utilization rate of code vectors in a codebook (or called dictionary) is not a novel problem, which has been early studied in dictionary learning.
>
> The contribution of our paper is to present an optimal VQ strategy tailored for the machine learning community, rather than merely addressing codebook collapse. In our paper, we examine two distinct issues in VQ: training stability and codebook collapse from the perspective of distributional matching, and empirically and theoretically demonstrate that identical distributions yield the optimal solution in VQ. Therefore, we not only achieve 100% codebook utilization, but also establish a stable VQ system with minimal information loss.

---

> ### Author Response · Authors · 2024-11-27
> **Response to reviewer YHdN (2/3)**
>
> **R 4.4** Is the low utilization rate of code vectors a result of the gradient backpropagation problem (as argued by the authors in line 049), or is it due to the insufficient diversity of training samples relative to the large size of the codebook? I lean towards the latter explanation, and therefore consider the low utilization rate is *not* a pivotal issue that significantly impacts the final performance of vector quantization.
>
> Thank you for your question. However, **we have never stated that the low utilization rate of code vectors is a result of the gradient backpropagation problem.** Instead, we clarify that our explanation addresses training instability, not the codebook collapse (low codebook utilization rate), which arises from the gradient backpropagation problem of copying the gradient of $z_q$ to $z_e$.
>
> While insufficient diversity of training samples might be correlated with the low codebook utilization, **the intrinsic issues of existing VQ methods should be the root cause**. For example, Vanilla VQ and k-means-based VQ algorithms partition the feature space into Voronoi cells by assigning each feature vector to the nearest code vector based on Euclidean distance. In this step, the absence of feature vectors assigning to certain code vectors results in the non-optimization and non-updating of these code vectors, as depicted in Figure 12 in Appendix H.3. In practical applications, particularly when the codebook size $K$ is substantial, a majority of code vectors remain unutilized and unmodified, leading to serious codebook collapse.
>
> Due to these intrinsic issues, Section 3.2 also revealed that the effectiveness of existing VQ methods relies heavily on **codebook initialization**. Specifically, when the codebook distribution is initialized as the feature distribution, each Voronoi cell is assigned feature vectors, making all code vectors are optimized or updated. However,  it would be profoundly challenging for initialized code vectors to comply with the feature distribution, as the feature distribution is unknown and changes dynamically during training in practical applications. To address this limitation, we propose an explicit distributional matching regularization approach, leveraging our novel quadratic Wasserstein distance approach.
>
> Lastly, the final performance of VQ is moderately influenced by the $\mathcal{U}$:
> - Consider two cases with identical codebook size $K$. In the one case with a higher utilization rate $\mathcal{U}$, this can be considered equivalent to employing a larger codebook size $K_1$, where $K_1<K$, assuming a 100% utilization rate; Conversely, another case with a lower utilization rate $\mathcal{U}$ can be equated to a smaller codebook size $K_2$, where $K_2<K_1$, again assuming a 100% utilization rate.
> - VQ functions as a compressor (from continuous latent space to discrete space), where minimal information loss indicates improved expressivity. Therefore, the final performance of VQ is highly related to quantization error $\mathcal{E}$, the smaller $\mathcal{E}$, the better VQ performance.
> - Let's consider optimal VQ scenarios where codebook distribution are closely aligned with feature distribution. As depicted in the Figure 4(a) ($\mu=0$) and 4(d) ($\sigma=1$), we can observed that the codebook size $K$ exerts a moderate influence  on $\mathcal{E}$ (see below Table).
>
>  | $K$ | 128 | 256 | 512 | 1024 | 2048 | 4096 | 8192 | 16384 |
> | --- | --- | --- | --- | --- | --- | --- | --- | --- |
> | $\mu=0$      | 33.31 | 31.23 | 29.47 | 27.73 | 26.19 | 24.69 | 23.36 | 22.14 |
> | $\sigma=1$ | 33.63 | 31.24 | 29.35 | 27.75 | 26.13 | 24.69 | 23.35 | 22.12 |
>
> Once again, we emphasize our contribution in achieving the minimal $\mathcal{E}$ by distributional matching.

---

> ### Author Response · Authors · 2024-11-27
> **Response to reviewer YHdN (3/3)**
>
> **R 4.5** In Lines 062-068, the authors should delve into the reasons behind the low code vector utilization rate observed in existing methods. For instance, in my view, the codebook initialization with k-means (Zhu et al. 2024) should work well in distribution matching, but performed worse in the paper. Could this be attributed to the issue of codebook utilization rate, or other factors, such as differences in network structures?
>
> First, we have discussed the low code vector utilization rate arises from **the intrinsic issues of existing VQ methods**. Additionally, as elucidated in Section 3.2, the inferior performance, including low code vector utilization, observed in existing vector quantization methods stems from their failure to effectively achieve distributional matching.
>
> Second,  we have run the VQGAN-LC code, and we observed that the k-means component in VQGAN-LC offers minimal utility, whereas the linear layer projection proves to be the genuinely effective aspect. When removing the linear layer projection in VQGAN-LC, it will suffer from serious codebook collapse as VQGAN.
>
> Finally, our superior image reconstruction performance stems not only from achieving 100% codebook utilization but also from the minimization of quantization error. Conversely, VQGAN-LC merely achieves close to 100% codebook utilization.
>
> [1] VQGAN-LC: Scaling the Codebook Size of VQGAN to 100,000 with a Utilization Rate of 99%
>
> **R 4.6** A codebook with a high utilization rate should closely match the distribution of feature vectors. Therefore, is it necessary to incorporate an additional distribution matching approach based on the Wasserstein distance?
>
> Thanks for your question. First, the statement **A codebook with a high utilization rate should closely match the distribution of feature vectors** is problematic. As depicted in Figure 3 (e) and (f), when code vectors are distributed within the range of feature vectors, but do not closely align with the distribution of feature vectors, the codebook can also attains a very high $\mathcal{U}$.
>
> Second, the quadratic Wasserstein distance, being the simplest statistical distance we can employ, boasts a minimal computational complexity, and its calculation incurs negligible additional time overhead during the training process. Other statistical distances, such as the Kullback-Leibler divergence or the Bhattacharyya Distance, are challenging to implement and incur substantial computational costs (please refer to  **R1.7** ). While incorporating additional distributional matching techniques might further enhance the matching quality, it would also introduce more complexity. Therefore, we do not recommend adding other distribution matching approaches due to their potential drawbacks.

---

> > ### Comment · Reviewer_YHdN · 2024-11-28
> >
> > Thank you for your response. It seems that my major concerns were not  addressed.
> >
> > **The rationality behind the Gaussian distribution assumption for feature vectors.** In R4.2, the authors  failed to address my concern directly. In practice, high-dimensional feature vectors often do not satisfy Gaussian distributions, rendering this assumption questionable. Why not consider a more realistic model, like a Gaussian mixture? Instead, the authors  leverage improved empirical results and other studies to validate their assumption. This  is logically flawed, since numerous factors can influence results. For example, the Wasserstein distance can  partly achieve  distribution matching,  even when data isn't Gaussian. Since the authors promote the Gaussian distribution-based matching method as a selling point, they should provide a straightforward proof, such as demonstrating the distribution through hypothesis testing. Otherwise, this selling point is unfounded.
> >
> > **The necessary of distribution matching.**    A codebook that is regulazried with high utilization rates  and **low quantization errors**  should approximate the distribution of feature vectors.  Considering  the higher complexity of the distribution matching method,  as commented by other reviewers, it may not be necessary to employ such technique.
> >
> > **The technical contribution of  improving code utilization rate.**  This technique has been developed in the area of dictionary learning. The contribution is incremental, rather than novel.

---

> ### Author Response · Authors · 2024-11-28
> **Discussion Round 1 with reviewer YHdN**
>
> Thank the reviewers for your prompt response. However, I would like to clarify several factural errors in reviewers' comments.
>
> - **Assuming of feature vectors in high-dimensional**. We acknowledge the challenge of applying Wasserstein distance in high-dimensional spaces, particularly when $d \geq 128$ (the popular Fréchet inception distance (FID) continues to assume Gaussian distribution within a very high-dimensional spaces.). However, within the specific context of VQ, the extracted feature vectors $z_e = E(x) \in \mathbb{R}^{h\times w \times d}$ from the encoder typically reside in a moderately large dimension space. e.g., $d=8$. Suppose $n$ image samples are used in one-step training; then, there are $n * h * w = 8192$ feature vectors for calculating the Wasserstein distance. Under the Multi-VQ setting, there are $n * h * w = 24576$ feature vectors in one training step for each GPUs. Under the condition that the number of feature vectors is sufficiently large while the dimensionality is sufficiently small, adopting the Gaussian assumption is not particularly problematic.
>
> -  **The authors promote the Gaussian distribution-based matching method as a selling point (this selling point is unfounded.)** Our main contribution is the empirical and theoretical demonstration, presented in Section 2, that **an identical distribution is the optimal solution for VQ** (we are the pioneers in presenting this optimal VQ insight to the machine learning community). We further elaborate on this in Section 3.2, where we detail how existing VQ methods such as $k$-means fall short in achieving this distribution matching.
>
> -  **They should provide a straightforward proof, such as demonstrating the distribution through hypothesis testing** We indeed included the theoretical proof in Section 2.4 but not the hypothesis testing the prove the identical distribution is the optimal solution to VQ.
>
> - **The higher complexity of the distribution matching method** Despite the seemingly complex formulation of the  quadratic  Wasserstein distance, its computational overhead is remarkably low, requiring only a single inversion of an 8-dimensional matrix ( by torch.linalg.eigh) followed by three matrix multiplication operations (torch.mm).
> $$
> \sqrt{\Vert \mu_1 - \mu_2 \Vert^2 + \text{tr}( {\Sigma_1}) + \text{tr}( {\Sigma_2})- 2\text{tr}( (\Sigma_1^{\frac{1}{2}} {\Sigma_2} \Sigma_1^{\frac{1}{2}})^{\frac{1}{2}})}  = \sqrt{\Vert \mu_1 - \mu_2 \Vert^2 + \text{tr}( {\Sigma_1}) + \text{tr}( {\Sigma_2})- 2\text{tr}( (\Sigma_1 \Sigma_2)^{\frac{1}{2}})}
> $$
>
> **R 4.7 the authors leverage improved empirical results and other studies to validate their assumption. This is logically flawed, since numerous factors can influence results.**
>
> We kindly request the reviewer to provide concrete evidence, rather than subjective evaluations. We would appreciate it if the reviewer could specify the exact issues in the corresponding sections and figures of the manuscript.
>
> **R 4.8 The technical contribution of improving code utilization rate. This technique has been developed in the area of dictionary learning. The contribution is incremental, rather than novel.**
>
> We have clarified this comment in **R 4.3**. **Our contribution is to provide an optimal VQ strategy tailored for the machine learning community, rather than merely addressing codebook collapse**. Could you identify any existing VQ paper that approaches the problem from the perspective of optimality?
>
> Finally, employing a Gaussian mixture model may provide a more accurate depiction of the distribution of feature vectors; nevertheless, it would augment the complexity of the entire distributional-matching methodology. This approach can be deemed as an extension for future research endeavors. We adopt the quadratic Wasserstein distance primarily for its simplicity and elegance.

---

> ### Comment · Reviewer_YHdN · 2024-11-28
>
> Thank you for your response.  Please see the comments:
>
> **Authors' Response 1:** They should provide a straightforward proof, such as demonstrating the distribution through hypothesis testing We indeed included the theoretical proof in Section 2.4 but not the hypothesis testing the prove the identical distribution is the optimal solution to VQ.
>
> **Comment:**  Regarding the "proof", what I mean is the evidence for the Gaussian distribution of feature vectors. If the feature distribution is not Gaussian, how do you ensure to achieve the perfect distribution matching (or, optimal VQ quantization)  using the  Wasserstein distance-based matching method?
>
> **Authors' Response 2:**  R 4.7 the authors leverage improved empirical results and other studies to validate their assumption. This is logically flawed, since numerous factors can influence results.----We kindly request the reviewer to provide concrete evidence, rather than subjective evaluations. We would appreciate it if the reviewer could specify the exact issues in the corresponding sections and figures of the manuscript.
>
> **Comment:** This question is also about the rationality of the Gaussian distribution assumption for feature vectors.
>
> **Authors' Response 3:**  We have clarified this comment in R 4.3. Our contribution is to provide an **optimal VQ** strategy tailored for the machine learning community, rather than merely addressing codebook collapse. Could you identify any existing VQ paper that approaches the problem from the perspective of optimality?
>
> **Comment:** The authors claimed that their major contribution is providing an **optimal** VQ strategy for the machine learning community.   If this claim holds true, the paper should progress in the following way. Initially, the authors should **mathematically** define what is  the **optimal VQ**, given a distribution of feature vectors. Subsequently, they must demonstrate that the VQ involved in deep networks can converge towards this **optimal VQ**, utilizing the Backpropagation (BP) algorithm under the constraint of Wasserstein distance-based distribution matching. Regrettably,  the two crucial elements are not found in Theorems 1 and 2.

---

> ### Author Response · Authors · 2024-11-28
> **Discussion Round 2 with reviewer YHdN**
>
> We thank the reviewers for their engagement in further discussion. We acknowledge the limitation of employing the Gaussian distribution assumption, as noted in Section 4.2. We recognize that our proposed Wasserstein distance-based matching method may not achieve precise density matching within deep neural networks, and we also cannot provide a theoretical proof for accurate density matching. Despite this, I would like clarify the following two points.
>
> First, we would provide our clarification on the **certain plausibility of employing the Gaussian distribution**.
>
> - We acknowledge that relying on the assumption of a Gaussian distribution may have undesired effects.  However, this assumption is commonly employed in various practices. For instance, the Fréchet inception distance (FID), a widely used metric for evaluating generated images from generative models like diffusion models, assumes that both the generated images and the ground-truth images follow a Gaussian distribution. Despite its limitations, FID remains a popular choice for measuring the distance between images in different domains. Additionally, the Gaussian distribution assumption is extensively utilized in numerous classical research works, including Variational Autoencoders (VAEs) [1], Generative Adversarial Networks (GANs) [2], Stable Diffusion (SD) [3], and Stable Diffusion XL (SDXL) [4].
> - The central limit theorem possibly plays a role in ensuring that learned feature vectors and code vectors follow a Gaussian distribution provided a large sample size and a large codebook size.
> - The real distribution of the feature and code vectors would be exceedingly intricate and often defies precise mathematical formulation. Our selection of the Gaussian assumption is primarily motivated by the Gaussian distribution’s inherent simplicity. Within the specific context of VQ, the feature vectors $z_e = E(x) \in \mathbb{R}^{h\times w \times d}$ extracted from the encoder typically reside in a moderately small dimension space ($d=8$). In Vanilla VQ setting, if $n$ image samples are used in one training step, there are $n * h * w = 8192$ feature vectors for calculating the Wasserstein distance. Under the Multi-VQ setting, each GPU processes $n * h * w = 24576$ feature vectors in a single training step. Given that the number of feature vectors is sufficiently large while the dimensionality is moderately small, adopting the Gaussian assumption is not particularly problematic and provides a reasonable approximation.
> - The effectiveness of the Gaussian assumption is also evident in our experimental results. For instance, in Section 3.2, despite initializing the codebook with a standard Gaussian distribution, the optimization process of the codebook distribution remains inherently arbitrary during training. Nevertheless, our quadratic Wasserstein distance, which is based on the Gaussian distribution assumption, consistently achieves superior VQ performance.
>
> Second, we would like to emphasize **the substantial challenges involved in attaining precise density matching.**
>
> - Achieving precise density matching is already a highly challenging problem in statistical theory. When considering its application in deep neural networks, where the feature distribution is entirely unknown, it becomes virtually an insurmountable task.
> - Representative research paper, such as Variational Autoencoders (VAEs) [1], utilize the Kullback-Leibler (KL) divergence to approximate two distributions. However, explicit evidence for precise density matching is not provided. Generative Adversarial Networks (GANs) [2] introduce an adversarial training framework to achieve density matching between generated and real images, while diffusion models[5] propose a multi-step denoising approach to accomplish the same task. Neither of these methods, however, offers a detailed explanation or theoretical proof of their density matching capabilities. All of these works are highly influential and extensively cited in the machine learning field.
>
> We currently fall short of achieving precise density matching. As the proverb goes, Rome was not built in a day. We are confident that, based on the insights we have provided, the pursuit of identical density matching between the feature distribution and the codebook distribution will inspire future research endeavors to achieve superior density matching, ultimately leading to enhanced VQ performance.
>
> [1] Auto-Encoding Variational Bayes, ICLR 2014
> [2] Generative Adversarial Nets, NeurIPS 2014
> [3] High-Resolution Image Synthesis with Latent Diffusion Models, CVPR 2022
> [4] SDXL: Improving Latent Diffusion Models for Hight-Resolution Image Synthesis
> [5] Denoising Diffusion Probabilistic Models

---

> ### Comment · Reviewer_YHdN · 2024-11-29
>
> Thank you for your response. Regrettably, the authors did not exactly grasp and answer my concerns. Please  carefully read my previous comments and **reply them one by one by directly referencing them**. This will help other readers to comprehend our  discussions. Below are my final comments.
>
>  **Comment 1: The Gaussian distribution assumption of  feature vectors.**   I understand that this assumption may be the best choice for the authors, especially concerning the adoption of the Wasserstein Distance. If the authors were to employ the more reasonable Gaussian mixtures assumption, it would increase the complexity in computation and analysis, thus hard to realize in a short time.
>
> In this context, my concern is the previous question:  "A codebook that is regulazried with high utilization rates and low quantization errors should approximate the distribution of feature vectors. Considering the higher complexity of the distribution matching method, as commented by other reviewers, it may not be necessary to employ such technique."   Regrettably, the authors have yet to provide direct and precise responses to each of my queries.
>
> **Comment 2: The major contribution: "optimal VQ".**  Here we first reference the response of the authors: "We have clarified this comment in R 4.3. Our contribution is to provide an optimal VQ strategy tailored for the machine learning community, rather than merely addressing codebook collapse. **Could you identify any existing VQ paper that approaches the problem from the perspective of optimality?**"
>
> **The answer is affirmative.   Numerous studies have delved into this issue.** "The history of optimal vector quantization theory dates back to the 1950s at Bell Laboratories, where researchers conducted studies to enhance signal transmission through suitable discretization techniques," as noted in the citation: Pagès G. Introduction to vector quantization and its applications for numerics[J]. ESAIM: proceedings and surveys, 2015, 48: 29-79.
>
> These studies  have investigated the optimal VQ issue from the traditional rate-distortion perspective, offering a more comprehensive and in-depth analysis compared to the theorems presented in the paper. This suggests that the innovation in this paper is quite limited.

---

> ### Author Response · Authors · 2024-11-29
> **Discussion Round 3 with reviewer YHdN**
>
> We thank the reviewers for their engagement in further discussion. We have addressed the comments from reviewers YHdN one-by-one. We sincerely hope that reviewers YHdN will carefully review our responses.
>
> **R 4.9 A codebook that is regulazried with high utilization rates and low quantization errors should approximate the distribution of feature vectors. Considering the higher complexity of the distribution matching method, as commented by other reviewers, it may not be necessary to employ such technique. Regrettably, the authors have yet to provide direct and precise responses to each of my queries.**
>
> We **have definitively provided** the direct and precise responses to the reviewers, and have categorized **The higher complexity of the distribution matching method** as factural errors. Please doule check **Discussion Round 1 with reviewer YHdN**.
>
> First, **the dimensionality of feature vectors is moderately small**. Within the context of VQ, the feature vectors extracted from the encoder, denoted as $z_e = E(x) \in \mathbb{R}^{h\times w \times d}$, typically reside in a space of modest dimensionality, e.g., $d=8$.
>
> Second, despite the seemingly complex formulation of the quadratic Wasserstein distance, **its computational overhead is remarkably low**, requiring only a single inversion of an 8-dimensional matrix ( by torch.linalg.eigh) followed by three matrix multiplication operations (torch.mm).
> $$
> \sqrt{\Vert \mu_1 - \mu_2 \Vert^2 + \text{tr}( {\Sigma_1}) + \text{tr}( {\Sigma_2})- 2\text{tr}( (\Sigma_1^{\frac{1}{2}} {\Sigma_2} \Sigma_1^{\frac{1}{2}})^{\frac{1}{2}})}  = \sqrt{\Vert \mu_1 - \mu_2 \Vert^2 + \text{tr}( {\Sigma_1}) + \text{tr}( {\Sigma_2})- 2\text{tr}( (\Sigma_1 \Sigma_2)^{\frac{1}{2}})}
> $$
> We strongly encourage the reviewers to read the Section 3.2 code in the supplementary material (wasserstein.py) for a comprehensive understanding.
>
>
> **R 4.10 The history of optimal vector quantization theory dates back to the 1950s at Bell Laboratories, where researchers conducted studies to enhance signal transmission through suitable discretization techniques," as noted in the citation: Pagès G. Introduction to vector quantization and its applications for numerics[J]. ESAIM: proceedings and surveys, 2015, 48: 29-79. These studies have investigated the optimal VQ issue from the traditional rate-distortion perspective, offering a more comprehensive and in-depth analysis compared to the theorems presented in the paper. This suggests that the innovation in this paper is quite limited.**
>
> We acknowledge that VQ is indeed studied by early research endeavors of the 20th century.  However, these studies on VQ were primarily confined to numerical computation and signal processing, e.g., [5] exemplified by the reviewer. We respectfully challenge the reviewer’s assessment of our contributions and the novelty of our innovations. We would like to clarify three points.
> - **In accordance with the reviewer’s assessment of novelty and contributions**, the seminal works on diffusion models [1][2] seem **quite limited**, given that the **diffusion process has been extensively studied** within the statistical community. Particularly, [2] employs score matching, a technique that was initially introduced in 2005 by [3]. We kindly ask the reviewer why this work was accepted as an oral presentation at NeurIPS 2019 and has garnered significant citations. Similarly, VQ has been extensively studied throughout the 20th century. It is our contention that VQ-VAE offers minimal novelty. We respectfully question why this paper is often regarded as the pioneering work in the field of visual generative models.
> - We respectfully challenge the reviewer’s statement that these studies have examined the optimal VQ issue from the traditional rate-distortion perspective. We kindly request the reviewer to specify the page and theorem in which this examination is detailed in [5].
> - We also respectfully challenge the reviewer’s statement that **these studies offer a more comprehensive and in-depth analysis compared to the theorems presented in the paper. the innovation in this paper is quite limited.** We kindly request the reviewers to explain what does the term “in-depth analysis” specifically refer to in [5]? What useful findings are there, and do the insights provided in G3 align with those offered in [5]?
>
> Based on the completely subjective judgment of Reviewer YHdN, we will approach the Program Chairs, Senior Area Chairs, and Area Chairs to address the reviewer’s lack of an objective and impartial reviewing stance.
>
> [1] DDPM: Denoising Diffusion Probabilistic Models NeurIPS 2020
> [2] Generative Modeling by Estimating Gradients of the Data Distribution, NeurIPS 2019
> [3] Estimation of Non-Normalized Statistical Models by Score Matching, JMLR 2005
> [4] Neural Discrete Representation Learning, NeurIPS 2017
> [5] Introduction to vector quantization and its applications for numerics

---

### Official Review · Reviewer_9xEZ · 2024-10-28

**Soundness:** 2
**Presentation:** 3
**Contribution:** 2
**Rating:** 3
**Confidence:** 4

**Summary:**

Existing vector quantization methods in autoregressive models suffer from training instability and codebook collapse due to a mismatch between feature and code vector distributions, causing significant data loss. To address the problem, the authors propose to use the Wasserstein distance to align these distributions, achieving high codebook utilization and reduced quantization error.

**Strengths:**

The Gaussian modeling-based distribution matching is proposed to enhance the codebook utilization rate, then improving the VQ performance.

**Weaknesses:**

As discussed in the paper, several methods have been introduced to enhance the codebook utilization rate. The paper's contribution appears to be more incremental than highly innovative. Additionally, the proposed distribution matching method simply assumes that all feature vectors adhere to a Gaussian distribution. This assumption may not be perfect for feature vectors of various patterns.

**Questions:**

1) There are simpler methods for matching  distributions, like the k-means, which does not require prior assumptions about the data distribution. This raises the question of whether  the Gaussian modeling-based distribution matching is  necessary, especially given its higher complexity.


2) Is Criterion 2 (Codebook utilization rate) necessary for optimization?  This criterion seems to be well satisfied if Criterion 3 (Codebook perplexity ) is satisfied.

3) The paper devotes considerable effort (in sections 2.3 and 2.4) to demonstrating the advantage of distribution matching (between feature vectors and code vectors) in enhancing quantization accuracy. This relationship is straightforward and readily understandable. However, the authors are encouraged to provide a more detailed examination of the convergence properties of the proposed three criteria.


4)  Does the multi-scale VQ cost (namely Eq. (5)) really work?  By 4.2, it is easy to derive that $z_q=z_1+(\hat{z}_T-z_T)$, implying that the features we adopt for codebook optimization essentially share the same resolution  with  $ z_T$, rather than multi-scale. For the operation on $z_i$, ($1<i<T$), the method implicitly assume that the feature vectors nearby in space are similar to each other, and then share the same quantization values (code vectors). However, this assumption is not appropriate for the low-resolution features with high variance.


5) In Figures 2 and 6, the sub-figures about  ${e_k}$  and ${\gamma}$ should be put in the rectangle of “vector quantizer”.

---

> ### Author Response · Authors · 2024-11-26
> **Response to reviewer 9xEZ (1/3)**
>
> **R 3.1** As discussed in the paper, several methods have been introduced to enhance the codebook utilization rate. The paper's contribution appears to be more incremental than highly innovative.
>
> Our paper aim to present an optimal VQ strategy tailored for the machine learning community, rather than merely enhancing the codebook utilization rate. By studying two distinct issues in VQ: training stability and codebook collapse from the perspective of distributional matching, we empirically and theoretically demonstrate that identical distributions yield the optimal solution in VQ. Although this finding is simple and intuitive, it is essential to emphasize that we are the pioneers in presenting this insight to the machine learning community.
>
> **R 3.2** There are simpler methods for matching distributions, like the k-means, which does not require prior assumptions about the data distribution. This raises the question of whether the Gaussian modeling-based distribution matching is necessary, especially given its higher complexity.
>
> We argue that $k$-means-based VQ methods are incapable of effectively matching distributions. The limitation of $k$-means-based VQ methods, such as VQ-VAE-2, notably the significant issue of codebook collapse, especially when the codebook size $K$ is large, is widely recognized in numerous research studies.
>
> To help reviewers better understand this limitation, we offer **visual illustrations to elucidate why $k$-means-based VQ methods fall short in achieving distributional matching**. As depicted in Figure 12 in Appendix H.3, $k$-means algorithm initially partitions the feature space into Voronoi cells by assigning each feature vector to the nearest code vector based on Euclidean distance. In this step, the absence of feature vectors assigning to certain code vectors results in the non-utilization and non-updating of these code vectors. In practical applications, particularly when the codebook size $K$ is substantial, a majority of code vectors remain unutilized and unmodified. This phenomenon underscores the inherent difficulty faced by $k$-means-based VQ methods in learning an effective and representative codebook for distributional matching.
>
> In addition, Section 3.2 presents an atomic and fair experimental setting where we empirically demonstrate that **Vanilla VQ and methods based on $k$-means and $k$-means++ fail to align the distributions of features and codebooks**. Within this setting, we initialize the codebook distribution as a standard Gaussian distribution across all VQ methods and treat code vectors as trainable parameters. Furthermore, we constrain the feature distribution across various VQ methods to a fixed Gaussian distribution by sampling feature vector $z_i  \sim \mathcal{N}(\mu \cdot 1_{m}, I_m)$. The performance under this setting is presented as follows:
>
>
> $k$-means (VQ-EMA)
> | Metrics\Means | 0 | 1 | 2 | 3 | 4 | 5 | 6 | 7 | 8 |
> | --- | --- | --- | --- | --- | --- | --- | --- | --- | --- |
> | $\mathcal{E}$ | 0.9816 | 1.0423 | 1.5325 | 2.2627 | 2.9776 | 3.5537 | 4.4314 | 4.4701 | 5.1897 |
> | $\mathcal{U}$  | 1.0000 | 0.7700 | 0.1442 | 0.0269 | 0.0081 | 0.0038 | 0.0015 | 0.0014 | 0.0007 |
> | $\mathcal{C}$  | 15171.5938 | 11757.9844 | 2277.2913 | 435.8142 | 132.2182 | 61.9389 | 23.9600 | 22.9621 | 11.9553 |
> | Wasserstein Distance | 0.0192 | 0.8813 | 5.2335 | 8.5722 | 11.4863 | 14.3154 | 17.1455 | 19.9417 | 22.7638 |
>
> Our Wasserstein VQ
> | Metrics\Means | 0 | 1 | 2 | 3 | 4 | 5 | 6 | 7 | 8 |
> | --- | --- | --- | --- | --- | --- | --- | --- | --- | --- |
> | $\mathcal{E}$  | 1.0527 | 1.0526 | 1.0523 | 1.0527 | 1.0524 | 1.0523 | 1.0527 | 1.0514 | 1.0523 |
> | $\mathcal{U}$  | 0.9996 | 0.9996 | 0.9997 | 0.9993 | 0.9996 | 0.9992 | 0.9992 | 0.9996 | 0.9992 |
> | $\mathcal{C}$  | 15132.9814 | 15117.4609 | 15132.0879 | 15146.6689 | 15121.4551 | 15156.2207 | 15151.8418 | 15135.4219 | 15128.8984 |
> | Wasserstein Distance | 0.0144 | 0.0168 | 0.0146 | 0.0147 | 0.0118 | 0.0147 | 0.0143 | 0.0124 | 0.0170 |
>
> It could be concluded that:
> - A small value of Wasserstein distance ensures superior VQ performances in terms of $(\mathcal{E}, \mathcal{U}, \mathcal{C})$;
> - $k$-means-based VQ methods fail to achieve distributional matching ($\mu \geq 2$), heavily relying on codebook initialization;
> - By using explicit distributional matching regularization, our proposed Wasserstein VQ is robust and independent of codebook initialization.
>
> We emphasize that it would be profoundly challenging for initialized code vectors to comply with the feature distribution, as the feature distribution is unknown and changes dynamically during training in practical applications.
>
> Our Section 3.2 code is available in the supplementary material. All VQ methods except Online clustering complete execution within five minutes. Within moderate high dimension, e.g., $d=8$, the computational cost incurred by the quadratic Wasserstein distance is virtually negligible.

---

> ### Author Response · Authors · 2024-11-26
> **Response to reviewer 9xEZ (2/3)**
>
> **R 3.3** Is Criterion 2 (Codebook utilization rate) necessary for optimization? This criterion seems to be well satisfied if Criterion 3 (Codebook perplexity ) is satisfied.
>
> Yes, Criterion 2 is essential. Criterion 2 assesses the completeness, whereas Criterion 3 evaluates the uniformity, which represent two distinct facets of codebook utilization. We have provided visual explanations in Appendix H.1 as to why Criterion 2 or Criterion 3 alone is insufficient to evaluate the extent of codebook collapse. As depicted in Figure 11 (a) and (d), despite identical $\mathcal{C}$, there is a significant difference in $\mathcal{U}$. Criterion 3 fails to account for the proportion of actively utilized code vectors in the VQ.
>
> **R 3.4** The paper devotes considerable effort (in sections 2.3 and 2.4) to demonstrating the advantage of distribution matching (between feature vectors and code vectors) in enhancing quantization accuracy. This relationship is straightforward and readily understandable. However, the authors are encouraged to provide a more detailed examination of the convergence properties of the proposed three criteria.
>
> Thanks for the reviewer’s suggestion. We are glad to add experimental examination of the convergence properties. However, at this moment, we are not fully clear which kind of experiments the reviewer are suggesting. Could you please illustrate a bit more about what kind of convergence properties need more examination? We believe we can add suitable experiments later.
>
> Additionally, although the finding that identical distributions yield the optimal solution in VQ is straightforward and intuitive, we are the first to present this insight to the machine learning community. It is imperative that we rigorously expound on this finding in the paper.
>
> **R 3.5 the proposed distribution matching method simply assumes that all feature vectors adhere to a Gaussian distribution. This assumption may not be perfect for feature vectors of various patterns.**
>
> We acknowledge that relying on the assumption of a Gaussian distribution may have undesired effects. However, it is worth noting that this assumption is commonly employed in various practices. For instance, the Fréchet inception distance (FID), a widely used metric for evaluating generated images from generative models like diffusion models, assumes that both the generated images and the ground-truth images follow a Gaussian distribution. Despite its limitations, FID remains a popular choice for measuring the distance between images in different domains.
>
> The real distribution of the feature and code vectors would be exceedingly intricate and often defies precise mathematical formulation. Our selection of the Gaussian assumption is primarily motivated by the Gaussian distribution’s inherent simplicity. Concurrently, the central limit theorem plays a role in ensuring that learned feature vectors and code vectors follow a Gaussian distribution provided a large sample size and a large codebook size. Indeed, the Gaussian assumption has been used in many studies such as in GANs, where the latent distribution is assumed to be Gaussian.
>
> The effectiveness of the Gaussian assumption is also evident in our experiments. For instance, in Section 3.2, despite initializing the codebook with a standard Gaussian distribution, the optimization process of the codebook distribution would inherently be arbitrary during training. Nevertheless, our quadratic Wasserstein distance metric—which is based on the Gaussian distribution assumption—still effectively aligns the distributions and achieves the best VQ performance.

---

> ### Author Response · Authors · 2024-11-26
> **Response to reviewer 9xEZ (3/3)**
>
> **R 3.6 Does the multi-scale VQ cost (namely Eq. (5)) really work? By 4.2, it is easy to derive that $z_q=z_1+(\hat{z}^T−z^T)$, implying that the features we adopt for codebook optimization essentially share the same resolution with $z^T$, rather than multi-scale. For the operation on $z_i$, ($1<i<T$), the method implicitly assume that the feature vectors nearby in space are similar to each other, and then share the same quantization values (code vectors). However, this assumption is not appropriate for the low-resolution features with high variance.**
>
> Yes, multi-scale VQ does indeed work, and we will provide empirical evidence later. We first explain that **codebook optimization is indeed multi-scale rather than the same resolution.** As shown in Block 3 of Figure 6, the feature vectors processed by vector quantizer are $g_i(z_i)$, which are produced by interpolating $z_i$ to a set of smaller resolutions (the resolution details can be found in Appendix F).
>
> Then, we explain why **multi-scale VQ does indeed work**.  According to $z_q=z_e+(\hat{z}^t−z^t)$ ($1 \leq t  \leq T$), we calculate the multi-scale quantization error $\mathcal{E}_t =\left\Vert  z_q-z_e \right\Vert_2^{2} = \left\Vert   \hat{z}^t− z^t \right\Vert_2^{2} =  \left\Vert  z^{t+1} \right\Vert_2^{2}$.  Interestingly, as presented in the table below, we can observed that $\mathcal{E}_t > \mathcal{E} _{t+1}, \forall 1 \leq t  \leq (T-1) $ in our image reconstruction experiments due to the convergence of the norm of  $z^{t+1}$.
>
> | Methods\t | 1 | 2 | 3 | 4 | 5 | 6 | 7 | 8 | 9 | 10|
> | --- | --- | --- | --- | --- | --- | --- | --- | --- | --- | ---|
> | Wasserstein VQ ($\alpha_3 = 0.0$) |  1.67  |  1.52  |  1.39   |  1.27   |  1.18  |  1.11  |  1.02  |  0.94  |  0.85  |   0.59  |
> | Wasserstein VQ ($\alpha_3 = 0.3$) |  0.29  |  0.25  |  0.21   |   0.17  |  0.15  |  0.13  |  0.11  |  0.09  |  0.08  |   0.03  |
>
> While you may notice that the incoporation of the quadratic Wasserstein distance can effectively reduce the quantization erorr from  $0.59$ to $0.03$, we don't report such improvement in Section 5, as we believe it is impossible to provide an atomic and fair setting to compare the quantization error in the image reconstruction experiments. Our analyses, as shown in Figures 8 and 9 in Appendix D, indicate that even when there is a matched distribution between code vectors and feature vectors, the quantization error can still exhibit considerable variability if the scales of those distributions differ. Therefore, we cannot confirm whether the quantization error improvement purely stems from the incorporation of the quadratic Wasserstein distance or the scale difference of feature distributions. The truly atomic and fair setting to compare different VQ methods have been introduced in Section 3.2.
>
> Finally, we are uncertain about the reviewer’s statement that **the method implicitly assume that the feature vectors nearby in space are similar to each other, and then share the same quantization values (code vectors). However, this assumption is not appropriate for the low-resolution features with high variance.** Could you please illustrate a bit more on that, so that we can engage in a more detailed discussion about this matter.

---

> ### Comment · Reviewer_9xEZ · 2024-11-28
>
> Thank you for providing these experiments to address my questions. The main limitation of the paper is its lack of substantial technical contribution. The paper aims to develop a codebook with uniform utilization rates to prevent codebook collapse. To achieve this, the authors introduce a distribution matching method based on the Wasserstein distance, which incorporates constraints on utilization rates and quantization errors. Statistically, simpler approaches such as online k-means and dictionary learning (corresponding to fully-connected structures) can also achieve distribution matching, while considering the  constraints mentioned above. In my view,  the constraints should play a more important role than  distribution matching.

---

> ### Author Response · Authors · 2024-11-28
> **Discussion Round 1 with Reviewer 9xEZ**
>
> Thank the reviewers for your prompt response, and would like to clarify two points.
>
> - **Contribution of this paper** We have clarified that **our contribution is to provide an optimal VQ strategy tailored for the machine learning community, rather than merely addressing codebook collapse** in **R 3.1**, Could you provide any existing VQ paper that approaches the problem from the perspective of optimality? Alternatively, do you believe that proposing an optimal VQ strategy is meaningless?
>
> - **$k$-means-based VQ methods fall short in achieving distributional matching** We have clarified this in **R 3.2** by visual explanation and empirical evidence. Could you provide any evidence to refute our viewpoint? Alternatively, if you have any points of confusion, please feel free to raise them, as we are more than willing to assist in clarifying this issue for the reviewers. We believe that productive discussions are grounded in evidence rather than subjective opinions.
>
> Let us know if you have any other questions or concerns. We again would like to thank you for your time and effort in reviewing our paper.

---

> > ### Comment · Reviewer_9xEZ · 2024-11-28
> >
> > Thank you for your clarification.  Please carefully read my comment:  "To achieve this, the authors introduce a distribution matching method based on the Wasserstein distance, which incorporates constraints on utilization rates and quantization errors. Statistically, simpler approaches such as online k-means and dictionary learning (corresponding to fully-connected structures) can also achieve distribution matching, while considering the constraints mentioned above. **In my view, the constraints should play a more important role than distribution matching.**
> >
> > In the experiments provided in R 3.2,  it seems that the k-means method  does not incorporate the constraints of  utilization rates and quantization errors. If the constraints, especially utilization rates, are introduced, I think the codebook collapse problem should be addressed.

---

> ### Author Response · Authors · 2024-11-28
> **Discussion Round 2 with Reviewer 9xEZ**
>
> Thank the reviewers for engaging in further discussion. I would like to clarify **two factural errors** in reviewers' comments.
>
> **Reviewers' Comment 1** The authors introduce a distribution matching method based on the Wasserstein distance, which incorporates constraints on utilization rates and quantization errors. the constraints should play a more important role than distribution matching.
>
> **Our Responses**: Our proposed quadratic Wasserstein distance **does not incorporates constraints on utilization rates and quantization errors.**
> $$
> \sqrt{\Vert \mu_1 - \mu_2 \Vert^2 + \text{tr}( {\Sigma_1}) + \text{tr}( {\Sigma_2})- 2\text{tr}( (\Sigma_1^{\frac{1}{2}} {\Sigma_2} \Sigma_1^{\frac{1}{2}})^{\frac{1}{2}})}
> $$
> **We are curious about why you think we have incorporated constraints like utilization rates and quantization errors? could you point out this in the above Equation?** Our proposed criteria, which comprises the codebook utilization rate $\mathcal{U}$, and quantization error $\mathcal{E}$ serves as an evaluation metric to substantiate the optimal VQ strategy in Section 2.3 and Section 2.4, as well as to illuminate the limitations of existing VQ methods in Section 3.2. It is imperative to highlight that **these metrics are not employed as losses**. **Without incorporation of any other constraints, the intrinsic effectiveness of distributional matching can achieve 100% codebook utilization rate and minimal quantization error**.  We strongly encourage the reviewers to read the Section 3.2 code in the supplementary material (wasserstein.py) for a comprehensive understanding.
>
> **Reviewers' Comment 2** it seems that the k-means method does not incorporate the constraints of utilization rates and quantization errors. If the constraints, especially utilization rates, are introduced, I think the codebook collapse problem should be addressed.
>
> **Our Responses**: First, we emphasize that the codebook utilization rate within Criterion 2 is non-differentiable. It merely serves as a statistic to measure VQ performance, and cannot be directly utilized as a loss function for optimization. **Could you elucidate how you can effectively incorporate the constraint of utilization rates into the k-means algorithm?**
>
> Second, **the k-means algorithm is intrinsically designed for the purpose of reducing quantization error**, please refer to the Appendix A.1 in [1]. We are also more than willing to clarify the distinction between Vanilla VQ-VAE and the $k$-means algorithm to the reviewers.
> - **Assignment step**:  Suppose there are a set of feature vectors $\\{ z_i\\} _{i=1}^N$ and code vectors $\\{e_k\\} _{k=1}^{K}$, in the $t$ assignment step, both two algorithms partition the feature space into Voronoi cells by assigning each feature vector to the nearest code vector based on Euclidean distance as follow:
> $$
> S_k^{(t)} = \\{z_i: \left\Vert  z_i-e_k^{(t)} \right\Vert_2^{2} \leq  \left\Vert  z_i-e_j^{(t)} \right\Vert_2^{2}, \forall j, 1\leq j \leq K\\}
> $$
> - **Update step in Vanilla VQ-VAE** It updates using gradient descent through the loss function provided below:
> $$
> \mathcal{L} = \frac{1}{N} \sum_{k=1}^{K} \sum_{m=1}^{|S_k^{(t)}|} \left\Vert z_m-e_k^{(t)} \right\Vert_2^{2}
> $$
> - **Update step in $k$-means** Recalculate means for feature vectors assigned to each code vectors:
> $$
> e_k^{t+1} = \frac{1}{|S_k^{(t)}|} \sum_{z_m \in S_k^{(t)}} z_m
> $$
> In practical implementations, the $k$-means algorithm employs a slightly more sophisticated approach, involving an exponential moving average of the assigned feature vectors. **The $k$-means algorithm was originally proposed to optimize quantization error.** However, this method inherently falls short of achieving minimal quantization error.
>
> [1] Neural Discrete Representation Learning, NeurIPS 2017
>
> Let us know if you have any other questions or concerns. We again would like to thank you for your time and effort in reviewing our paper.

---

> > ### Author Response · Authors · 2024-11-30
> > **rebuttal discussion**
> >
> > Dear Reviewer 9xEZ,
> >
> > We recognize that the timing of this discussion period may not align perfectly with your schedule, yet we would greatly value the opportunity to continue our dialogue before the deadline approaches.
> >
> > Could you let us know if your questions have been adequately addressed? If not, please feel free to raise them, and we are more than willing to provide further clarification; if you find that your concerns have been resolved, we would appreciate if you could re-consider the review score.
> >
> > We hope that we have resolved all your questions, but please let us know if there is anything more.
> >
> > Best wishes to you!

---

### Official Review · Reviewer_TvXJ · 2024-11-01

**Soundness:** 2
**Presentation:** 2
**Contribution:** 2
**Rating:** 5
**Confidence:** 3

**Summary:**

The paper tackles key issues in vector quantization (VQ) within autoregressive models, specifically addressing training instability and codebook collapse. The authors identify these issues as a mismatch between the distributions of features and code vectors, leading to significant information loss during compression. They propose aligning the codebook distribution with the feature distribution using quadratic Wasserstein distance to improve three criteria they introduce—Quantization Error, Codebook Utilization Rate, and Codebook Perplexity. The paper validates the effectiveness of these metrics through empirical analysis and further substitutes vanilla VQ with a multi-scale VQ structure. Experimental results demonstrate near 100% codebook utilization, reduced quantization error, and performance improvements.

**Strengths:**

1. The paper thoroughly analyzes the issues with VQ, particularly the training instability due to gradient discontinuities (addressed via the Straight-Through Estimator, though still prone to instability), and the low codebook utilization due to limited active code vectors.
2. The introduction of three specific criteria—Quantization Error, Codebook Utilization Rate, and Codebook Perplexity—adds quantifiable methods for assessing VQ performance and measuring improvements objectively.
3. The alignment of the feature distribution with the codebook distribution via quadratic Wasserstein distance is well-motivated theoretically and effectively addresses the identified issues.
4. The paper validates its proposed approach through experiments, showing substantial improvements in codebook utilization (nearly 100%) and other metrics, supporting the contributions' significance.
5. The multi-scale VQ approach, combined with Wasserstein distance loss, shows promise for practical applications, especially in visual generative tasks.

**Weaknesses:**

1. While Gaussian sampling effectively validates the proposed metrics, testing on a broader range of data distributions would enhance the generalizability of the findings, particularly for distributions commonly encountered in autoregressive models or visual tasks. Notably, VQ-VAE does not assume a Gaussian distribution in its latent space, making evaluating performance across diverse data types essential to reflect real-world applications better.
2. Although the metrics (Quantization Error, Codebook Utilization Rate, and Codebook Perplexity) are proposed and demonstrated through Gaussian sampling, the paper lacks validation on diverse and realistic data, especially those commonly used in autoregressive and generative tasks. The reliance on Gaussian sampling limits the generalizability and applicability of the metrics, making it uncertain how they would perform in varied contexts.
3. The paper's focus on aligning feature and codebook distributions with Wasserstein distance for improved quantization seems limited in scope and impact. The primary advantage is codebook utilization. However, the broader implications or potential applications in complex or large-scale generative tasks (e.g., real-world autoregressive models) are not clearly demonstrated.
4. While the paper achieves nearly 100% codebook utilization, the practical significance of this improvement remains unclear. It's difficult to determine if the proposed method advances the field without a comprehensive comparison to other quantization methods that might achieve similar outcomes or more straightforward approaches to training stabilization.
5. While multi-scale VQ improves vanilla VQ, the paper lacks detailed theoretical grounding or motivation for why a multi-scale approach is advantageous, specifically in Wasserstein distance.

**Questions:**

1. Could you clarify the theoretical motivation behind using multi-scale VQ over vanilla VQ and its role in improving Wasserstein alignment?
2. Have you tested your metrics on real-world datasets beyond Gaussian sampling to validate their general applicability?
3. Can you clarify the practical implications of 100% codebook utilization, especially when compared to alternative methods in vector quantization?

---

> ### Author Response · Authors · 2024-11-27
> **Response to reviewer TvXJ (1/3)**
>
> **R 2.1**  While Gaussian sampling effectively validates the proposed metrics, testing on a broader range of data distributions would enhance the generalizability of the findings, particularly for distributions commonly encountered in autoregressive models or visual tasks. Notably, VQ-VAE does not assume a Gaussian distribution in its latent space, making evaluating performance across diverse data types essential to reflect real-world applications better.
>
> Although VQ-VAE does not assume a Gaussian distribution in its latent space, VQ-VAE has many shortcomings as well:
> - **Severe codebook collapse** VQ-VAE partition the feature space into Voronoi cells by assigning each feature vector to the nearest code vector based on Euclidean distance. In this step, the absence of feature vectors assigning to certain code vectors results in the non-optimization and non-updating of these code vectors, as depicted in Figure 12 in Appendix H.3. In practical applications, particularly when the codebook size is substantial, a majority of code vectors remain unutilized and unmodified, leading to serious codebook collapse.
> - **Relies heavily on codebook initialization** In Section 3.2, we reveal that the effectiveness of VQ-VAE relies heavily on codebook initialization. Specifically, when the codebook distribution is initialized as the feature distribution, each Voronoi cell is assigned feature vectors, making all code vectors are optimized or updated. However, it would be profoundly challenging for initialized code vectors to comply with the feature distribution, as the feature distribution is unknown and changes dynamically during training in practical applications. To address this limitation, we propose an explicit distributional matching regularization approach, leveraging our novel quadratic Wasserstein distance approach.
> - **Training instability** In the absence of proper codebook initialization, VQ-VAE will encounter a very large $\mathcal{E}$, resulting in unstable training.
>
> In contrast, our proposed Wasserstein VQ does not suffer from the above shortcomings. Also, in our experience, Gaussian assumption is not too restrictive. Indeed, it is commonly assumed in various works, such as in GAN and self-supervised learning, that the learned representations follow a Gaussian distribution. The central limit theorem plays a role in guaranteeing this provided a large sample size.  Moreover, this assumption has also been heavily used in evaluation. For instance, the Fréchet inception distance (FID), a widely used metric for evaluating generated images from generative models like diffusion models, assumes that both the generated images and the ground-truth images follow a Gaussian distribution. Despite its limitations, FID remains a popular choice for mea suring the distance between images in different domains.
>
>
> **R 2.2**  Although the metrics (Quantization Error, Codebook Utilization Rate, and Codebook Perplexity) are proposed and demonstrated through Gaussian sampling, the paper lacks validation on diverse and realistic data, especially those commonly used in autoregressive and generative tasks. The reliance on Gaussian sampling limits the generalizability and applicability of the metrics, making it uncertain how they would perform in varied contexts.
>
> Although relying on Gaussian assumption, we also demonstrate the effectiveness of the quadratic Wasserstein distance when code vectors and feature vectors are arbitrary distributions during training, in Section 3.2 and Section 5, respectively. Given that image data always have a bounded support, we do not think the Gaussian assumption is a critical assumption. Moreover, it is commonly assumed in various works, such as in GAN and self-supervised learning, that the learned representations follow a Gaussian distribution.

---

> ### Author Response · Authors · 2024-11-27
> **Response to reviewer TvXJ (2/3)**
>
> **R 2.3** The paper's focus on aligning feature and codebook distributions with Wasserstein distance for improved quantization seems limited in scope and impact. The primary advantage is codebook utilization. However, the broader implications or potential applications in complex or large-scale generative tasks (e.g., real-world autoregressive models) are not clearly demonstrated.
>
> Thanks for this suggestion. Due to limited compute resource, we are unable to demonstrate potential applications in complex or large-scale generative tasks. However, we are able to show the Wasserstein loss does improve reconstruction performance on FFHQ and Imagenet dataset. Also, the primary advantage of Wasserstein VQ does not merely lie in codebook utilization rate; it also achieves the best performance in terms of quantization error among other VQ approaches. We provide evidence of this in Section 3.2 when code vectors are arbitrary distributions during training.
>
> **R 2.4** While the paper achieves nearly 100% codebook utilization, the practical significance of this improvement remains unclear. It's difficult to determine if the proposed method advances the field without a comprehensive comparison to other quantization methods that might achieve similar outcomes or more straightforward approaches to training stabilization.
>
> We argue that **we not only achieve nearly 100% codebook utilization but also attain the minimal quantization error**. As discussed in **R 1.8**, it would be almost impossible to compare the performance of various VQ methods in terms of the criterion triple in image reconstruction experiments, due to the lack of a truly atomic and fair experimental setting for such comparisons.
>
> Specifically, the distribution of feature vectors produced by the encoder varies across different VQ methods, which complicates the establishment of a fair comparison of quantization error. Our analyses, as shown in Figures 8 and 9 in Appendix D, indicate that even when there is a matched distribution between code vectors and feature vectors, **the quantization error can still exhibit considerable variability if the scales of those distributions differ**. Since the feature distribution is unknown and dynamically changes during training—and evolves uniquely for each VQ method—it is difficult to maintain the same feature distribution scales across various methods. As a result, such comparisons are neither atomic nor fair. This is the reason we did not include a comparison of different VQ methods in terms of quantization error in Section 5.
>
> In **R 3.6**, the incoporation of the quadratic Wasserstein distance is shown to effectively reduce the quantization erorr from $0.59$ to $0.03$. However, we consider it is not an atomic and fair setting to compare the quantization error in the image reconstruction experiments as we cannot confirm whether the quantization error improvement purely stems from the incorporation of the quadratic Wasserstein distance or the scale difference of feature distributions. Therefore, we don't report such improvement in Section 5.
>
> In Section 3.2, we provide a simple yet atomic experimental setting by configuring feature distribution across various VQ methods to a fixed Gaussian distribution. While feature distributions are generally intricate and dynamic in practical application scenarios, this simplified setting still yields valuable insights, as indicated in **G3**  within the general response to all reviewers. One notable insight is that our proposed Wasserstein VQ  achieves near-optimal quantization error without relying on codebook initialization, whereas the effectiveness of existing VQ methods is heavily contingent upon the ideal codebook initialization. Moreover, achieving an ideal codebook initialization that accurately mirrors the feature distribution is practically unattainable due to the unknown and dynamically evolving nature of feature distributions during training in real-world applications.

---

> ### Author Response · Authors · 2024-11-28
> **Response to reviewer TvXJ (3/3)**
>
> **R 2.5** Have you tested your metrics on real-world datasets beyond Gaussian sampling to validate their general applicability?
>
> As discussed in **R 2.4**, experiments conducted on real-world datasets fall short of providing a fair setting for the comparison of various VQ methods based on our proposed metrics. In Section 3.2, we establish a simple yet fair experimental setting to assess diverse VQ methods using our proposed evaluation criteria.
>
> **R 2.6** Can you clarify the practical implications of 100% codebook utilization, especially when compared to alternative methods in vector quantization?
>
> First, VQ functions as a compressor (from continuous latent space to discrete space), where minimal information loss indicates improved expressivity. Therefore, the final performance of VQ is highly related to quantization error $\mathcal{E}$, **the smaller $\mathcal{E}$, the better VQ performance.**
>
> Second, we can explain **the quantization error is moderately influenced by the $\mathcal{U}$**.
> - Consider two cases with identical codebook size $K$. In the one case with a higher utilization rate $\mathcal{U}$, this can be considered equivalent to employing a larger codebook size $K_1$, where $K_1<K$, assuming a 100% utilization rate; Conversely, another case with a lower utilization rate $\mathcal{U}$ can be equated to a smaller codebook size $K_2$, where $K_2<K_1$, again assuming a 100% utilization rate.
> - Let's consider optimal VQ scenarios where codebook distribution are closely aligned with feature distribution, and codebook utilization is 100%. As depicted in the Figure 4(a) ($\mu=0$) and 4(d) ($\sigma=1$), we can observed that the codebook size $K$ exerts a moderate influence  on $\mathcal{E}$ (see below Table).
>
>  | $K$ | 128 | 256 | 512 | 1024 | 2048 | 4096 | 8192 | 16384 |
> | --- | --- | --- | --- | --- | --- | --- | --- | --- |
> | $\mu=0$      | 33.31 | 31.23 | 29.47 | 27.73 | 26.19 | 24.69 | 23.36 | 22.14 |
> | $\sigma=1$ | 33.63 | 31.24 | 29.35 | 27.75 | 26.13 | 24.69 | 23.35 | 22.12 |
>
> Therefore, 100% codebook utilization can ensures reduced quantization error by the larger $K$.
>
> Once again, we emphasize that **our approach to achieving minimal $\mathcal{E}$ is not solely dependent on the 100% codebook utilization rate, but is instead primarily facilitated by the distributional matching**. This distinction is pivotal, as a 100% codebook utilization rate, on its own, does not guarantee minimal quantization error. As depicted in Figure 3 (e) and (f), when code vectors are distributed within the range of feature vectors, but do not closely align with the distribution of feature vectors, the codebook can also attains a very high $\mathcal{U}$ while yielding a significantly larger $\mathcal{E}$.

---

> > ### Author Response · Authors · 2024-11-30
> > **rebuttal discussion**
> >
> > Dear Reviewer TvXJ,
> >
> > We recognize that the timing of this discussion period may not align perfectly with your schedule, yet we would greatly value the opportunity to continue our dialogue before the deadline approaches.
> >
> > Could you let us know if your questions have been adequately addressed? If not, please feel free to raise them, and we are more than willing to provide further clarification; if you find that your concerns have been resolved, we would appreciate if you could re-consider the review score.
> >
> > We hope that we have resolved all your questions, but please let us know if there is anything more.
> >
> > Best wishes to you!

---

> > > ### Comment · Reviewer_TvXJ · 2024-12-02
> > >
> > > Thank you for the response. While you addressed some of my concerns, several issues remain unresolved. Therefore, I would like to keep my score as it is.

---

### Official Review · Reviewer_bT9x · 2024-11-05

**Soundness:** 2
**Presentation:** 2
**Contribution:** 3
**Rating:** 6
**Confidence:** 4

**Summary:**

To address the training instability and codebook collapse in vector quantization, the authors align the distribution of features and code vectors by including an additional quadratic Wasserstein distance term which regularizes their disparity under a Gaussian assumption. This method leads to 100% codebook utilization and better image reconstruction/generation quality.

**Strengths:**

1. The Wasserstein VQ algorithm is efficient and has a mathematical foundation
1. Empirical experiments indicate improvement in reconstruction quality

**Weaknesses:**

1. The presentation is not clear enough. See the questions below, some of which are crucial.
1. The improvement in reconstruction quality is marginal. The authors also didn't explain how they picked the additional hyperparameter $\alpha_3$, which can raise suspicion of $p$-hacking.
1. Insufficient ablation experiments. The authors should study how the choice of distance measure (Wasserstein vs KL divergence), $\alpha_3$, and other knobs affect the performance of the proposed method.
1. Editorial:
    1. On line 93, it's inaccurate to say "codeword IDs $r \in \mathbb{R}^{h \times w}$", because the components in $r$ are discrete.
    1. When presenting the experiment in Figure 3, the authors should point the reader to Appendix E.1, which includes details of the experimental setting.
    1. On line 280, consider calling the optimal set of code vectors $\{e_K^*\}_{k=1}^K$ to avoid confusion.
    1. There is an extra "Then" in Lemma 3 on line 310.
    1. In Section 4.1 on line 370, it's clearer to use the existing notations $z_i$ and $e_i$, instead of defining new ones $z_e$ and $z_q$.
    1. In Section 4.2 on line 418, the footnote $ ^3$ looks like a cubic power notation.
    1. In Section 6 on line 534, there is a typo in "alignment".

**Questions:**

1. The authors mentioned training instability on line 107, but it's unclear to me how Wasserstein VQ can help stabilize the learning. Is the argument that a smaller VQ reconstruction error makes $z_i$ and $e_i$ closer in expectation, and thus corresponds to better learning stability?
1. For "Criterion 2 (Codebook Utilization Rate)" defined on line 156, how does it depend on $N$?
1. For "Criterion 3 (Codebook Perplexity)" defined on line 180, do we want a high codebook perplexity? I am a little confused because in language modeling we want the "perplexity" to be low.
1. The interpretation of Figure 3 in lines 203 through 213 is not deep enough. In particular, why do $\mathcal{E}$, $\mathcal{U}$, and $\mathcal{C}$ exhibit different behavior when the feature covers the codebook vs vice versa?
1. In Section 2.4, the theorems all require bounded support, but Section 3.1 has the Gaussian assumption, while Gaussian does not have a bounded support. It's fine to use the theorems as an intuitive argument, but the authors should at least explain why their theoretical results "somehow" apply to Gaussian when the assumption is violated.
1. What's the motivation for the Gaussian assumption in Section 3.1? What's the real distribution of the features and code vectors in the experiments?
1. On line 307, it is claimed that KL divergence between Gaussian distributions has no analytical formula, but there is one on [Wikipedia](https://en.wikipedia.org/wiki/Kullback%E2%80%93Leibler_divergence#Multivariate_normal_distributions). It would be great if the authors can check if that is correct and explain why the quadratic Wasserstein distance is a better measure than the KL divergence.
1. In Section 3.2 and Figure 5, it is unclear what "training steps" mean. By inspection, we can see Equation (3) has an optimal solution at $\hat{\mu}_2^* = \hat{\mu}_1$ and $\hat{\Sigma}_2^* = \hat{\Sigma}_1$. We can then transform the code vectors such that $\hat{\mu}_2 = \hat{\mu}_2^*$ and $\hat{\Sigma}_2 = \hat{\Sigma}_2^*$ with a linear transformation. I don't think there is anything to be trained.
1. In Section 3.2 on line 346, what's the motivation for sampling the code vectors such that $\hat{\mu}_2 \ne \hat{\mu}_1$, which is clearly suboptimal? Moreover, how does Wasserstein VQ compare against other methods when $\hat{\mu}_2 = \hat{\mu}_1$ at initialization?

---

> ### Author Response · Authors · 2024-11-25
> **Response to reviewer bT9x (1/4)**
>
> Our code is available in the supplementary material, all experimental results are reproducible, and we promise that our code will be made publicly available. Due to the high computational demands of training—each parameter adjustment requires 8 H20 GPUs and takes two days—we were unable to conduct a parameter sensitivity analysis due to limited funding and GPU resources. We selected $\alpha=0.3$ based on loss scaling and extensive tuning experience. This aligns with common practice in the machine learning community, where experiments requiring substantial computational resources typically skip parameter sensitivity analyses.
>
> **R 1.1** The authors mentioned training instability on line 107, but it's unclear to me how Wasserstein VQ can help stabilize the learning. Is the argument that a smaller VQ reconstruction error makes $z_i$ and $e_i$ closer in expectation, and thus corresponds to better learning stability?
>
> Yes, Wasserstein VQ effectively minimizes the quantization error between $z_q$ and $z_e$ in expectation, thereby reducing the gradient gap between them. This reduction in gradient gap helps stabilize the training. Indeed, we observe training instability when the gradient gap is large. This is possibly due to the use of STE and a large gradient gap, the training of the codebook is unstable as sometimes the closest code vectors change during training.  Additionally, VQ functions as a compressor (from continuous latent space to discrete space), where minimal information loss indicates improved expressivity. Our proposed Wasserstein VQ makes the information loss minimal.
>
> **R 1.2** For "Criterion 2 (Codebook Utilization Rate)" defined on line 156, how does it depend on N.
>
> Criterion 2 has a direct relationship with sample size N. Here's how this relationship works:
>
> 1. When the sample size is smaller than the codebook size, 100% $\mathcal{U}$ is impossible—the pigeonhole principle dictates that some code vectors will lack corresponding feature vectors.
> 2. When the sample size equals half the codebook size, $\mathcal{U}$ can reach a maximum of 50%.
> 3. Only when the sample size significantly exceeds the codebook size can $\mathcal{U}$ approach 100%.
>
> Our empirical evidence in Figures 4(i), 4(l), 7(i), and 7(l) confirms this pattern: even with identical data distributions, small sample sizes prevent full codebook utilization. In the paper, we define Criterion 2 using a finite set of feature vectors and code vectors. When considering Criterion 2 for a feature distribution and a set of code vectors, $\mathcal{U}$ remains unknown, but we can analyze it asymptotically by setting the sample size $N \to \infty$:
>
> $$
> \begin{aligned}
> \mathcal{U} (\{e_k\}; \{z_i\})=\lim _{N \to \infty} \frac{1}{N}\sum _{i=1}^N 1 _{(e_k =z_i' \textnormal{ for some }i)}
> \end{aligned}
> $$
>
> **R 1.3** For "Criterion 3 (Codebook Perplexity)" defined on line 180, do we want a high codebook perplexity? I am a little confused because in language modeling we want the "perplexity" to be low.
>
> Yes, high codebook perplexity is indeed what we desire, even though codebook perplexity shares the same mathematical formula as in language modeling. In vector quantization (VQ), code vector utilization $p_k$ is analogous to sub-word frequencies in a text corpus. When codebook perplexity reaches its minimum, only one code vector is being used, indicating severe codebook collapse—similar to having a text corpus with only one subword, which is clearly undesirable. While subword frequencies in natural language follow specific patterns, in visual generative tasks we expect uniform usage of all code vectors without any vision-specific prior, thus leading to maximum codebook perplexity.
>
> This differs from the perplexity use in the language modeling, where perplexity measures model performance rather than word frequency distribution. For instance, given "the movie is [MASK]," we expect specific evaluative words like "good" or "bad" rather than random selections from the entire dictionary. We also provide the comprehensive understanding of Criterion 2 and 3 in the Appendix H.1.

---

> ### Author Response · Authors · 2024-11-25
> **Response to reviewer bT9x (2/4)**
>
> **R 1.4** The interpretation of Figure 3 in lines 203 through 213 is not deep enough. In particular, why do $\mathcal{E}$, $\mathcal{U}$ , and $\mathcal{C}$ exhibit different behavior when the feature covers the codebook vs vice versa?
>
> Thank you for your insightful question. The VQ process relies on the nearest neighbor search for code vector selection. When code vectors are distributed within the range of feature vectors, as illustrated in Figure 3 (e) and (f), the majority of code vectors would be actively utilized, ensuring high $\mathcal{U}$. However, the utilization of these code vectors is not uniform; code vectors on the periphery of the codebook distribution are more frequently used, leading to relatively low $\mathcal{C}$. Feature vectors on the periphery will have larger distances to their nearest code vectors, resulting in higher $\mathcal{E}$. Conversely, when feature vectors fall within the range of code vectors, as depicted in Figure 3 (g) and (h), outer code vectors remain largely unused, leading to a lower $\mathcal{U}$ and $\mathcal{C}$. Since only inner code vectors are active, each feature vector can find a nearby counterpart, maintaining low $\mathcal{E}$. More detailed explanations can be found in Appendix H.2
>
> **R 1.5** In Section 2.4, the theorems all require bounded support, but Section 3.1 has the Gaussian assumption, while Gaussian does not have a bounded support. It's fine to use the theorems as an intuitive argument, but the authors should at least explain why their theoretical results "somehow" apply to Gaussian when the assumption is violated.
>
> Thank you for raising this theoretical question. We would like to answer this question from two perspectives. First, we would like to note that Theorem 2 does not really need the assumption of bounded support. We have made revisions on the presentation of Theorem 2, removing the boundedness assumption. Thus, this theorem will encompass Gaussian distributions as special examples. Second, for Theorem 1, the boundedness of the support is necessary because we use the worst case quantization error in Equation 2. When the support is unbounded, the worst case quantization error will remain at $\infty$. However, in real applications when $\\{z_i\\} _{i=1}^N$ are generated from an absolutely continuous distribution in $\mathbb{R}^d$, these samples will be bounded with high probability. Therefore, our Theorem 1 would still provide insights for distributions with unbounded supports.
>
> **R 1.6** What's the motivation for the Gaussian assumption in Section 3.1? What's the real distribution of the features and code vectors in the experiments?
>
> The real distribution of the feature and code vectors would be exceedingly intricate and often defies precise mathematical formulation. Our selection of the Gaussian assumption is primarily motivated by the Gaussian distribution’s inherent simplicity. Concurrently, the central limit theorem plays a role in ensuring that learned feature vectors and code vectors follow a Gaussian distribution provided a large sample size and a large codebook size. Indeed, the Gaussian assumption has been used in many studies such as in GANs, where the latent distribution is assumed to be Gaussian.
>
> We acknowledge that relying on the assumption of a Gaussian distribution may have undesired effects. However, it is worth noting that this assumption is commonly employed in various practices. For instance, the Fréchet inception distance (FID), a widely used metric for evaluating generated images from generative models like diffusion models, assumes that both the generated images and the ground-truth images follow a Gaussian distribution. Despite its limitations, FID remains a popular choice for measuring the distance between images in different domains.
>
> The effectiveness of  the Gaussian assumption is also evident in our experiments. For instance, in Section 3.2, despite initializing the codebook with a standard Gaussian distribution, the optimization process of the codebook distribution would inherently be arbitrary during training. Nevertheless, our quadratic Wasserstein distance metric—which is based on the Gaussian distribution assumption—still effectively aligns the distributions and achieves the best VQ performance.

---

> ### Author Response · Authors · 2024-11-25
> **Response to reviewer bT9x (3/4)**
>
> **R 1.7**  On line 307, it is claimed that KL divergence between Gaussian distributions has no analytical formula, but there is one on Wikipedia. It would be great if the authors can check if that is correct and explain why the quadratic Wasserstein distance is a better measure than the KL divergence.
>
> Thank you for pointing this out. It should be read as “ … lack **simple** closed-form representations, making them computationally expensive.” Suppose two random variables $Z_1 \sim \mathcal{N}(\mu_1, \Sigma_1)$ and $Z_2 \sim \mathcal{N}(\mu_2, \Sigma_2)$ obey multivariate normal distributions, then Kullback-Leibler divergence between $Z1$ and $Z_2$ is:
> $$
> \mathcal{KL}(Z_1, Z_2) = \frac{1}{2}((\mu_1-\mu_2)^T \Sigma_2^{-1}(\mu_1-\mu_2) + \text{tr}(\Sigma_2^{-1} \Sigma_1- I) + \ln{\frac{\det{\Sigma_2}}{\det \Sigma_1}}).
> $$
> It can be observed that the KL divergence for two Gaussian distributions involves calculating the determinant of covariance matrices, which is computationally expensive in moderate and high dimensions. Moreover, the calculation of the determinant is sensitive to perturbations and it requires full rank (In the case of not full rank, the determinant is zero, rendering the logarithm of zero undefined), which can be impractical in many cases. Other statistical distances like Bhattacharyya Distance suffer from the same issue. In contrast, quadratic Wasserstein distance does not require the calculation of the determinant and full-rank covariance matrices.
>
> **R 1.8** In Section 3.2 and Figure 5, it is unclear what "training steps" mean. By inspection, we can see Equation (3) has an optimal solution at $\hat{\mu}_2^{*}=\hat{\mu}_1$ and $\hat{\Sigma}_2^ * =\hat{\Sigma}_1$. We can then transform the code vectors such that  $\hat{\mu}_2=\hat{\mu}_2^ *$ and $\hat{\Sigma}_2=\hat{\Sigma}_2^ *$   with a linear transformation. I don't think there is anything to be trained.
>
> The reviewer’s question likely stems from a misunderstanding of the experimental setting and motivation detailed in Section 3.2.  In the revised version of Section 3.2, we provide the clear motivation of our experiments. We greatly encourage the reviewer to revisit the modified content of Section 3.2, and we welcome the reviewer to raise any new questions.
>
> It would be extremely challenging to compare the performance of VQ methods in terms of the criterion triple in image reconstruction experiments, due to the lack of a truly atomic and fair experimental setting for such comparisons. Specifically, the distribution of feature vectors produced by the encoder varies across different VQ methods, which complicates the establishment of a fair comparison of quantization error. Our analyses, as shown in Figures 8 and 9 in Appendix D, indicate that even when there is a matched distribution between code vectors and feature vectors, the quantization error can still exhibit considerable variability if the scales of those distributions differ.  Since the feature distribution is unknown and dynamically changes during training—and evolves uniquely for each VQ method—it is difficult to maintain the same feature distribution scales across various methods. As a result, such comparisons are neither atomic nor fair. This is the reason we did not include a comparison of different VQ methods in terms of quantization error in Section 5. This challenge also encourages us to develop a truly atomic and fair setting to compare different methods, which will be the focus of Section 3.2.
>
> In Section 3.2, we provide a simple and atomic experimental setting by setting feature distribution in various VQ methods to a fixed Gaussian distribution. While feature distributions are typically complex and dynamic in practical application scenarios, this simplified setting still yields valuable insights, please refer to **G3** in the general response to all reviewers.  We treat the initialized code vectors as trainable parameters. The concept of **training step** in this experiment is the same as that in Section 5.  For example, in Vanilla VQ, **training step** denotes the times of updation of code vector by the SGD optimizer, while in VQ EMA, **training step** represents the times of updation of code vector by  $k$-means.
>
> We set the difference between the codebook distribution and the feature distribution because the feature distribution is unknown and dynamically adjusted in practical applications, rendering it impossible to directly initialize the codebook distribution using the feature distribution, i.e., $\mu_2 = \mu_1$. Additionally, we are curious to examine whether existing VQ methods have the capability to learn code vectors that match the distribution of feature vectors, given the unknown nature of the feature distribution.

---

> ### Author Response · Authors · 2024-11-25
> **Response to reviewer bT9x (4/4)**
>
> **R1.9** In Section 3.2 on line 346, what's the motivation for sampling the code vectors such that $\hat{\mu}_2 \neq \hat{\mu}_2$, which is clearly suboptimal? Moreover, how does Wasserstein VQ compare against other methods when $\hat{\mu}_2 = \hat{\mu}_1$ at initialization?
>
> We have elucidated the motivation behind sampling the code vectors such that $\hat{\mu}_2 \neq \hat{\mu}_2$ in **R 1.8**.  Initializing  $\hat{\mu}_2 = \hat{\mu}_1$ is essentially meaningless for comprehending VQ methods in real-world applications.
>
> **R 1.10** Insufficient ablation experiments. The authors should study how the choice of distance measure (Wasserstein vs KL divergence) and other knobs affect the performance of the proposed method.
>
> First, we have discussed the advantage of quadratic Wasserstein distance over KL divergence in **R 1.7**.
>
> Second, the primary focus of this paper is to present an optimal VQ strategy tailored for the machine learning community, rather than merely focusing on distributional matching through statistical distances. Consequently, our primary objective is to demonstrate the optimality of identical distribution alignment, while the implementation of this alignment is considered a secondary objective.  Although the finding that identical distributions yield the optimal solution in VQ is straightforward and intuitive, it is crucial to underscore that we are the pioneers in presenting this insight to the machine learning community. Therefore,  the explication of why identical distributions achieve optimality in VQ should be prioritized as a primary objective.
>
> Lastly, while we acknowledge the value of supplementing our study with experiments involving other statistical methods for distributional alignment, we fall short of GPU resources in this rebuttal period.
>
> **R 1.11**  In Section 4.1 on line 370, it's clearer to use the existing notations $z_i$ and $e_i$, instead of defining new ones $z_e$ and $z_q$.
>
> We have defined $z_e$ and $z_q$ earlier in Section 2.1. We also thank the reviewer for pointing out several editorial errors and assisting us in improving the manuscript.

---

> > ### Comment · Reviewer_bT9x · 2024-11-29
> >
> > I thank the authors for their detailed rebuttal, which partially addresses my concerns. I am not entirely convinced that if the Gaussian assumption is reasonable (this should be easy to check by collecting $z$'s and plotting the histogram of their norm), and ultimately, distributional matching only brings marginal improvement in reconstruction quality. However, this paper did address the training stability and codebook collapse, and can potentially serve as a stepping stone for subsequent research. Moreover, I'm glad to see that the authors relax the assumption of bounded support, slightly closing the gap between theory and reality. For these reasons, I'm increasing my rating.

---

> ### Author Response · Authors · 2024-11-29
> **Rebuttal discussion**
>
> Dear Reviewers bT9x,
>
> We greatly appreciate the reviewer’s objective and fair comments. The several editorial errors pointed out by the reviewers, along with the misunderstandings regarding **R 1.8** and **R 1.9**, have brought to our attention certain writing issues, and thus indeed improving the manuscript.
>
> **Reviewers' Comment 1**: distributional matching only brings marginal improvement in reconstruction quality.
>
> We are highly appreciative of the reviewer’s keen insight in identifying this aspect, which strongly affirms that the reviewers are indeed senior researchers. **The primary reason for the issues is that our baseline code contains certain deficiencies.**
>
> Initially, our intention was to conduct experiments based on VAR’s code. However, VAR has not fully open-sourced their VQVAE code, particularly the components related to GAN. Consequently, we opted to use the VQGAN-LC codebase. While VQGAN-LC has made its code publicly available, **its GAN training has consistently failed to converge to an equilibrium solution**. The training of the GAN is critical for enhancing the rFID metric; without proper GAN training, the generated images tend to be overly smooth. Our low rFID score is primarily attributable to the non-convergence of GAN training. After the submission deadline, we began adjusting the GAN based on LLamaGen’s code[1] (rFID reported in LLamaGen is 2.19), and our current rFID score has improved to approximately 0.9. Due to limited GPU resources, our new experiments results remain incomplete, and therefore, we have not included these updated results in the revised version of our manuscript.
>
> **Reviewers' Comment 2**:  I am not entirely convinced that if the Gaussian assumption is reasonable (this should be easy to check by collecting z's and plotting the histogram of their norm)
>
> Upon submission, we did not anticipate the extensive questioning of the Gaussian assumption by the reviewers. This assumption is widely utilized in machine learning and is a fundamental component of classical models such as Variational Autoencoders (VAEs), Generative Adversarial Networks (GANs), diffusion models, and stable diffusion models, all of which rely on Gaussian distributions. As a result, we underestimated the importance of providing a detailed justification for the Gaussian distribution assumption.
>
> In our rebuttal, we failed to consider the indirect method suggested by the reviewer for addressing this issue. Specifically, we were at a loss regarding how to visualize the density in an 8-dimensional space. We are deeply grateful for the reviewer’s insightful suggestions and will carefully incorporate them into our future work.
>
> [1] Autoregressive Model Beats Diffusion: Llama for Scalable Image Generation

---

### Author Response · Authors · 2024-11-26
**General commention to all reviewers**

Thank you to all the reviewers for your valuable feedback. In response to the reviewers’ feedback, we have made major revisions to the paper, summarized as follows:

- **Distributional matching analyses of existing VQ methods**  We have developed an atomic experimental setting to examine these VQ approaches through the lens of distributional matching, which is detailed in Section 3.2.
- **Visual explanations of the two criteria** To enhance clarity and understanding, we have incorporated visual elucidations to illustrate Criterion 2 and Criterion 3, as shown in Figure 11 of Appendix H.1.
- **Visual illustrations of the limitation of $k$-means based VQ approach** We offer visual illustrations to elucidate why k-means-based VQ methods fall short in achieving distributional matching, as depicted in Figure 12 in Appendix H.3.
- **Improved theoretical guarantees and discussions.** We improve our Theorem 2 by removing the assumption of bounded support, making it applicable to Gaussian distributions. In addition, we explicitly discuss the implication of Theorem 1.

Moreover, we acknowledge certain misunderstandings in the reviewer’s comments and are willing to clarify the following three points.

**G1 The main contribution of this paper lies in addressing codebook collapse or low codebook utilization rate**

Our paper aims to present an optimal VQ strategy tailored for the machine learning community, rather than merely addressing codebook collapse. In our paper, we examine two distinct issues in VQ: training stability and codebook collapse from the perspective of distributional matching, and empirically and theoretically demonstrate that identical distributions yield the optimal solution in VQ. Therefore, we not only achieve 100% codebook utilization, but also establish a stable VQ system with minimal information loss.

We also emphasize that we are the pioneers in presenting this optimal VQ solution to the machine learning community, despite its simplicity and intuitiveness.

**G2 $k$-means-based VQ methods can also achieve distributional matching.**

The limitation of $k$-means-based VQ methods, notably the significant issue of codebook collapse, especially when the codebook size $K$ is large, is widely recognized in numerous research studies. To help reviewers better understand this limitation, we offer visual illustrations to elucidate why
$k$-means-based VQ methods fall short in achieving distributional matching in Appendix H.3. Furthermore, in Section 3.2,  we have developed an atomic experimental setting to demonstrate that Vanilla VQ and methods based on $k$-means and $k$-means++ fail to align the distributions of features and codebooks. Additionally, we show that without explicit distributional matching regularization, the effectiveness of the existing VQ methods heavily depends on codebook initialization, which can become problematic when the feature distribution is unknown and dynamically changing during training.

**G3 The motivation of experimental setting and several insights in Section 3.2**

We recognize that it is nearly impossible to provide an atomically fair experimental setup to compare the performance of various VQ methods based on the criterion triple in image reconstruction experiments, as thoroughly discussed in **R 1.8.** This challenge motivates us to establish a genuinely atomic and fair comparison setting, which is the focal point of Section 3.2. Within this setting, we constrain the feature distribution across different VQ methods to a fixed Gaussian distribution. Although feature distributions are usually intricate and dynamic in real-world applications, this simplified setting yields valuable insights.

- Vanilla VQ and methods based on $k$-means and $k$-means++ demonstrate inferior performance due to their inability to align the distributions of features and codebooks, as evidenced by the experimental results depicted in Figure 5.
- In the absence of explicit distributional matching regularization, the effectiveness of existing VQ methods is significantly contingent upon codebook initialization. Conversely, with the incorporation of explicit distributional matching regularization, the performance of our proposed Wasserstein VQ is robust and independent of codebook initialization.
- Despite the inherent arbitrariness of the codebook distribution during training, our proposed quadratic Wasserstein distance, based on the Gaussian distribution assumption, efficiently aligns the distributions and attains superior VQ performance.

Our code is available in the supplementary material, and we have added the experimental code for Section 3.2. All experimental results are reproducible, and we promise that our code will be made publicly available. Once again, we appreciate the reviewers’ thoughtful comments, and we have carefully addressed each of them in our revised paper.

---

### Author Response · Authors · 2024-11-28
**Looking Forward to Deeper Discussions with All Reviewers**

Dear Area Chairs and All Reviewers,

Thank you to all the reviewers for your valuable feedback. We have dedicated more than two weeks to carefully address the comments of each reviewer, by providing a comprehensive 14-page response that includes 33 detailed answers. We recognize that not all reviewers may fully grasp the motivation and contributions of our paper, and thus, we have provided extensive explanations to address every concern raised. If any further concerns arise, please do not hesitate to bring them forth; we are eager to engage in deeper discussions with all reviewers.

Best Regards,
1683 Authors

---

### Author Response · Authors · 2024-11-29
**Request for Public Comment: Reviewer YHdN Request Us To Provide a Theoretical Proof for Accurate Density Matching In Deep Neural Networks**

Dear Program Chairs, Senior Area Chairs, Area Chairs, All Reviewers, and All Readers,

In our paper, we empirically and theoretically demonstrate that an identical distribution between feature vectors and code vectors yields the optimal solution in Vector Quantization (VQ). To achieve this distributional matching, we propose a quadratic Wasserstein distance metric based on the Gaussian distribution hypothesis. Notably, we are the pioneers in presenting this optimal VQ insight to the machine learning community.

However, **Reviewers YHdN** contends that the actual feature distribution is not Gaussian, and thus argues that our paper suffers from logical flaws, deeming our contribution as incremental. Furthermore, they request us to provide a theoretical proof for accurate density matching using the Wasserstein distance.

First, we clarify **the substantial challenges involved in precise density matching.**

- Achieving precise density matching is already a highly challenging problem in statistical theory. When considering its application in deep neural networks, where the feature distribution is entirely unknown, it becomes virtually an insurmountable task.
- Representative research paper, such as Variational Autoencoders (VAEs) [1], utilize the Kullback-Leibler (KL) divergence to approximate two distributions. However, explicit evidence for precise density matching is not provided. Generative Adversarial Networks (GANs) [2] introduce an adversarial training framework to achieve density matching between generated and real images, while diffusion models[5] propose a multi-step denoising approach to accomplish the same task. Neither of these methods, however, offers a detailed explanation or theoretical proof of their density matching capabilities. All of these works are highly influential and extensively cited in the machine learning field.

Second, we further clarify **certain plausibility of employing the Gaussian distribution**.

- We acknowledge that relying on the assumption of a Gaussian distribution may have undesired effects.  However, this assumption is commonly employed in various practices. For instance, the Fréchet inception distance (FID), a widely used metric for evaluating generated images from generative models, assumes that both the generated images and the ground-truth images follow a Gaussian distribution. Despite its limitations, FID remains a popular choice for measuring the distance between images in different domains. Additionally, the Gaussian distribution assumption is extensively utilized in numerous classical research works, including Variational Autoencoders (VAEs) [1], Generative Adversarial Networks (GANs) [2], Stable Diffusion (SD) [3], and Stable Diffusion XL (SDXL) [4].
- The central limit theorem possibly plays a role in ensuring that learned feature vectors and code vectors follow a Gaussian distribution provided a large sample size and a large codebook size.
- The real distribution of the feature and code vectors would be exceedingly intricate and often defies precise mathematical formulation. Our selection of the Gaussian assumption is primarily motivated by the Gaussian distribution’s inherent simplicity. Within the specific context of VQ, the feature vectors $z_e = E(x) \in \mathbb{R}^{h\times w \times d}$ extracted from the encoder typically reside in a moderately small dimension space (d=8). In Vanilla VQ setting, if $n$ image samples are used in one training step, there are $n * h * w = 8192$ feature vectors for calculating the Wasserstein distance. Under the Multi-VQ setting, each GPU processes $n * h * w = 24576$ feature vectors in a single training step. Given that the number of feature vectors is sufficiently large while the dimensionality is moderately small, adopting the Gaussian assumption is not particularly problematic and provides a reasonable approximation.
- The effectiveness of the Gaussian assumption is also evident in our experimental results. For instance, in Section 3.2, despite initializing the codebook with a standard Gaussian distribution, the optimization process of the codebook distribution remains inherently arbitrary during training. Nevertheless, our quadratic Wasserstein distance, which is based on the Gaussian distribution assumption, consistently achieves superior VQ performance.

[1] Auto-Encoding Variational Bayes, ICLR 2014
[2] Generative Adversarial Nets, NeurIPS 2014
[3] High-Resolution Image Synthesis with Latent Diffusion Models, CVPR 2022
[4] SDXL: Improving Latent Diffusion Models for Hight-Resolution Image Synthesis
[5] Denoising Diffusion Probabilistic Models

We sincerely appreciate the time and effort each reader has dedicated to reviewing our work. We strongly encourage readers to provide their comments and offer their perspectives on the reasonableness of the reviewers’ assessments and requests. Your insights are invaluable to us as we strive for continuous improvement in our research.

Best Regards,
1683 Authors

---

> ### Comment · Reviewer_YHdN · 2024-11-29
>
> Thank you for your response. Regrettably, the authors did not exactly grasp and answer my concerns. Please  carefully read my previous comments and **reply them one by one by directly referencing them**. This will help other readers to comprehend our  discussions. Below are my final comments.
>
>  **Comment 1: The Gaussian distribution assumption of  feature vectors.**   I understand that this assumption may be the best choice for the authors, especially concerning the adoption of the Wasserstein Distance. If the authors were to employ the more reasonable Gaussian mixtures assumption, it would increase the complexity in computation and analysis, thus hard to realize in a short time.
>
> In this context, my concern is the previous question:  "A codebook that is regulazried with high utilization rates and low quantization errors should approximate the distribution of feature vectors. Considering the higher complexity of the distribution matching method, as commented by other reviewers, it may not be necessary to employ such technique."   Regrettably, the authors have yet to provide direct and precise responses to each of my queries.
>
> **Comment 2: The major contribution: "optimal VQ".**  Here we first reference the response of the authors: "We have clarified this comment in R 4.3. Our contribution is to provide an optimal VQ strategy tailored for the machine learning community, rather than merely addressing codebook collapse. **Could you identify any existing VQ paper that approaches the problem from the perspective of optimality?**"
>
> **The answer is affirmative.  Numerous studies have delved into this issue.** "The history of optimal vector quantization theory dates back to the 1950s at Bell Laboratories, where researchers conducted studies to enhance signal transmission through suitable discretization techniques," as noted in the citation: Pagès G. Introduction to vector quantization and its applications for numerics[J]. ESAIM: proceedings and surveys, 2015, 48: 29-79.
>
> These studies  have investigated the optimal VQ issue from the traditional rate-distortion perspective, offering a more comprehensive and in-depth analysis compared to the theorems presented in the paper.  This suggests that the innovation in this paper is quite limited.

---

> > ### Comment · Reviewer_9xEZ · 2024-12-02
> >
> > From the above discussion in comment 2, I observed that the authors' claimed major contribution, the first study of optimal Vector Quantization (VQ), has been extensively researched, a fact seemingly overlooked by the authors. In their response, the authors did not explain their advantages or novelty over existing optimal VQ studies, a necessary clarification they should have provided. This makes the major contribution ambiguous.

---

> ### Author Response · Authors · 2024-11-29
> **Official Comment by 1683 Authors**
>
> We thank the reviewers for their engagement in further discussion. We have addressed the comments from reviewers YHdN one-by-one. We sincerely hope that reviewers YHdN will carefully review our responses.
>
> **Reviewers' Comment**: A codebook that is regulazried with high utilization rates and low quantization errors should approximate the distribution of feature vectors. Considering the higher complexity of the distribution matching method, as commented by other reviewers, it may not be necessary to employ such technique. Regrettably, the authors have yet to provide direct and precise responses to each of my queries.
>
> We **have definitively provided** the direct and precise responses to the reviewers, and have categorized **The higher complexity of the distribution matching method** as factural errors. Please doule check **Discussion Round 1 with reviewer YHdN**.
>
> First, **the dimensionality of feature vectors is moderately small**. Within the context of VQ, the feature vectors extracted from the encoder, denoted as $z_e = E(x) \in \mathbb{R}^{h\times w \times d}$, typically reside in a space of modest dimensionality, e.g., $d=8$.
>
> Second, despite the seemingly complex formulation of the quadratic Wasserstein distance, **its computational overhead is remarkably low**, requiring only a single inversion of an 8-dimensional matrix ( by torch.linalg.eigh) followed by three matrix multiplication operations (torch.mm).
> $$
> \sqrt{\Vert \mu_1 - \mu_2 \Vert^2 + \text{tr}( {\Sigma_1}) + \text{tr}( {\Sigma_2})- 2\text{tr}( (\Sigma_1^{\frac{1}{2}} {\Sigma_2} \Sigma_1^{\frac{1}{2}})^{\frac{1}{2}})}  = \sqrt{\Vert \mu_1 - \mu_2 \Vert^2 + \text{tr}( {\Sigma_1}) + \text{tr}( {\Sigma_2})- 2\text{tr}( (\Sigma_1 \Sigma_2)^{\frac{1}{2}})}
> $$
>
> **Reviewers' Comment** The history of optimal vector quantization theory dates back to the 1950s at Bell Laboratories, where researchers conducted studies to enhance signal transmission through suitable discretization techniques," as noted in the citation: Pagès G. Introduction to vector quantization and its applications for numerics[J]. ESAIM: proceedings and surveys, 2015, 48: 29-79. These studies have investigated the optimal VQ issue from the traditional rate-distortion perspective, offering a more comprehensive and in-depth analysis compared to the theorems presented in the paper. This suggests that the innovation in this paper is quite limited.
>
> We acknowledge that VQ is indeed studied by early research endeavors of the 20th century.  However, these studies on VQ were primarily confined to numerical computation and signal processing, e.g., [5] exemplified by the reviewer. In the context of deep learning tasks, the pioneering work is the VQ-VAE. We respectfully challenge the reviewer’s assessment of our contributions and the novelty of our innovations. We would like to clarify three points.
> - **In accordance with the reviewer’s assessment of novelty and contributions**, the seminal works on diffusion models [1][2] seem **quite limited**, given that the **diffusion process has been extensively studied** within the statistical community. Particularly, [2] employs score matching, a technique that was initially introduced in 2005 by [3]. We kindly ask the reviewer why this work was accepted as an oral presentation at NeurIPS 2019 and has garnered significant citations. Similarly, VQ has been extensively studied throughout the 20th century. It is our contention that VQ-VAE offers minimal novelty. We respectfully question why this paper is often regarded as the pioneering work in the field of visual generative models.
> - We respectfully challenge the reviewer’s statement that these studies have examined the optimal VQ issue from the traditional rate-distortion perspective. We kindly request the reviewer to specify the page and theorem in which this examination is detailed in [5].
> - We also respectfully challenge the reviewer’s statement that **these studies offer a more comprehensive and in-depth analysis compared to the theorems presented in the paper. the innovation in this paper is quite limited.** We kindly request the reviewers to explain what does the term “in-depth analysis” specifically refer to in [5]? What useful findings are there, and do the insights provided in **G3** align with those offered in [5]?
>
> [1] DDPM: Denoising Diffusion Probabilistic Models NeurIPS 2020
> [2] Generative Modeling by Estimating Gradients of the Data Distribution, NeurIPS 2019
> [3] Estimation of Non-Normalized Statistical Models by Score Matching, JMLR 2005
> [4] Neural Discrete Representation Learning, NeurIPS 2017
> [5] Introduction to vector quantization and its applications for numerics

---

> ### Author Response · Authors · 2024-12-02
> **Discussion to Reviewer 9xEZ**
>
> Dear Reviewer 9xEZ,
>
> Thanks for your discussion. Early studies on VQ were primarily confined to numerical computation and signal processing, e.g., [1] exemplified by the reviewer YHdN. Could you cite any papers that study the optimality of Vector Quantization (VQ), and indicate whether these studies have been widely applied to visual tokenizers? show your evidence rather than subjective comments. I am very grateful that ICLR is an open platform, where all comments will be visible even if our paper is rejected. We believe that the Program Chairs, Senior Area Chairs, Area Chairs will handle such instances of malicious rejection with due seriousness and rigor.
>
> [1]  Introduction to vector quantization and its applications for numerics.
>
> Best
> 1683 Authors

---

### Author Response · Authors · 2024-11-29
**1683 Authors Publicly Question the Objectivity and Impartiality of Reviewer YHdN’s Comments (1/2)**

Dear Program Chairs, Senior Area Chairs, Area Chairs, All Reviewers, and All Readers,

Leveraging the transparency of the OpenReview platform, 1683 authors have publicly questioned the objectivity and impartiality of Reviewer YHdN’s comments, citing their subjective perspectives and several factual errors.

We commence by summarizing the **discussion concerning novelty and contribution**.

- **Reviewer Comment**: Improving the utilization rate of code vectors in a codebook (or called dictionary) is not a novel problem, which has been early studied in dictionary learning.
- **Our Response**: The contribution of our paper is to present an optimal VQ strategy tailored for the machine learning community, rather than merely addressing codebook collapse, as detailed in **R 4.3**.
- **Round 1 Reviewer Comment**: The technical contribution of improving code utilization rate. This technique has been developed in the area of dictionary learning. The contribution is incremental, rather than novel.
- **Our Round 1 Response**: We have clarified this comment in **R 4.3**. Our contribution is to provide an optimal VQ strategy tailored for the machine learning community, rather than merely addressing codebook collapse. Could you identify any existing VQ paper that approaches the problem from the perspective of optimality?
- **Round 2 Reviewer Comment** The authors claimed that their major contribution is providing an optimal VQ strategy for the machine learning community. If this claim holds true, the paper should progress in the following way. Initially, the authors should mathematically define what is the optimal VQ, given a distribution of feature vectors. Subsequently, they must demonstrate that the VQ involved in deep networks can converge towards this optimal VQ, utilizing the BP algorithm under the constraint of Wasserstein distance-based distribution matching. Regrettably, the two crucial elements are not found in Theorems 1 and 2.
- **Our Round 2 Response** **1)** In Section2, based on two critical issues of VQ, we introduce a criterion triple that an optimal VQ must satisfy. We then empirically and theoretically demonstrate that an identical distribution between feature vectors and code vectors yields the optimal solution in VQ. Thus, we provide **a mathematically rigorous definition of optimal VQ**.  **2)** In Section 3.1, we propose the Wasserstein distance to achieve distributional matching under the Gaussian distribution assumption. While we acknowledge the difficulty in providing a theoretical proof for optimal solutions in deep neural networks due to the NP-hard nature of accurate density matching, we note that influential works such as VAEs use KL divergence andGANs use adversarial training to approximate or match distributions. Similarly, diffusion models employ multi-step denoising techniques. None of these methods, however, offer explicit evidence or a detailed theoretical proof of their density matching capabilities **3)** The reviewer **was unable to find any existing VQ papers** that address the problem from an optimality perspective and raised demands that the authors consider unreasonable. As the saying goes, Rome was not built in a day. We are confident that our insights will inspire future research to refine and enhance density matching, ultimately leading to improved VQ performance.
- **The Round 3 Reviewer Comment**  The history of optimal VQ theory dates back to the 1950s at Bell Laboratories, where researchers conducted studies to enhance signal transmission through suitable discretization techniques[4]. These studies have investigated the optimal VQ issue from the traditional rate-distortion perspective, offering a more comprehensive and in-depth analysis compared to the theorems presented in the paper. This suggests that the innovation in this paper is quite limited.
- **Our Round 3 Response** **1)** Early studies on VQ were primarily confined to numerical computation and signal processing, e.g., [4] by the reviewer. In the context of deep learning tasks, the pioneering work is the VQ-VAE  **2)** By the reviewer’s assessment of novelty and contributions, the seminal works on diffusion models [1][2] appear quite limited, considering the extensive study of diffusion processes within the statistical community. Specifically, [2] employs score matching, a technique introduced in 2005 by [3]. We kindly ask the reviewer why this work was accepted as an oral presentation at NeurIPS 2019 and has received significant citations. Similarly, VQ has been extensively studied throughout the 20th century, then why VQVAE is often regarded as the pioneering work in the field of visual generative models.

[1] DDPM: Denoising Diffusion Probabilistic Models NeurIPS 2020 [2] Generative Modeling by Estimating Gradients of the Data Distribution, NeurIPS 2019 [3] Estimation of Non-Normalized Statistical Models by Score Matching, JMLR 2005  [4] Introduction to vector quantization and its applications for numerics

---

> ### Author Response · Authors · 2024-11-29
> **1683 Authors Publicly Question the Objectivity and Impartiality of Reviewer YHdN’s Comments (2/2)**
>
> Dear Program Chairs, Senior Area Chairs, Area Chairs, All Reviewers, and All Readers,
>
> Then, we would like to list several factual errors in Reviewers' comments:
> -  **Reviewer Summary** In the paper, the authors highlighted that a low utilization rate of the codebook will impair the training stability of Vector Quantization (VQ)
> - **Our Response** Thank you for summarizing our paper. However, it is important to clarify that **we have never claimed that** a low utilization rate of the codebook impairs the training stability of VQ. (Explanations see in **4.1**)
> - **Reviewer Comment** Is the low utilization rate of code vectors a result of the gradient backpropagation problem (as argued by the authors in line 049), or is it due to the insufficient diversity of training samples relative to the large size of the codebook? I lean towards the latter explanation, and therefore consider the low utilization rate is not a pivotal issue that significantly impacts the final performance of vector quantization.
> - **Our Response** (1) See in **R4.4**, **we have never stated** that the low utilization rate of code vectors is a result of the gradient backpropagation problem. The true reason for the low utilization rate of code vectors is that Voronoi cell partitions cannot guarantee that all code vectors will be assigned feature vectors. High utilization rates of Voronoi cell partitions require close matching between the codebook distribution and the feature distribution. However, existing VQ algorithms struggle to ensure distributional matching during the training process in Section 3.2. (2) We also explain why the final performance of VQ is moderately influenced by the codebook utilization in **R4.4**.
> - **Reviewer Comment**  In my view, the codebook initialization with k-means (Zhu et al. 2024) should work well in distribution matching, but performed worse in the paper.
> - **Our Response** See in **R 4.5**, we have discussed the low code vector utilization rate arises from **the intrinsic issues** of existing VQ methods in **R 4.4**. Additionally, as elucidated in Section 3.2, the inferior performance, including low code vector utilization, observed in existing VQ methods stems from their failure to effectively achieve distributional matching.
> - **Reviewer Comment** High-dimensional feature vectors often do not satisfy Gaussian distributions, rendering this assumption questionable.
> - **Our Response** We acknowledge the challenge of applying Wasserstein distance in high-dimensional spaces, particularly when $d \geq 128$ (the popular Fréchet inception distance (FID) continues to assume Gaussian distribution within a very high-dimensional spaces.). However, within the specific context of VQ, the extracted feature vectors $z_e = E(x) \in \mathbb{R}^{h\times w \times d}$ from the encoder typically reside in a moderately large dimension space. e.g., $d=8$. Suppose $n$ image samples are used in one-step training; then, there are $n * h * w = 8192$ feature vectors for calculating the Wasserstein distance. Under the Multi-VQ setting, there are $n * h * w = 24576$ feature vectors in one training step for each GPUs. Under the condition that the number of feature vectors is sufficiently large while the dimensionality is sufficiently small, adopting the Gaussian assumption is not particularly problematic.
> - **Reviewer Comment** A codebook that is regulazried with high utilization rates and low quantization errors should approximate the distribution of feature vectors. Considering the higher complexity of the distribution matching method, as commented by other reviewers, it may not be necessary to employ such technique.
> - **Our Response** Despite the seemingly complex formulation of the quadratic Wasserstein distance, **its computational overhead is remarkably low**, requiring only a single inversion of an 8-dimensional matrix ( by torch.linalg.eigh) followed by three matrix multiplication operations (torch.mm).
> $$
> \sqrt{\Vert \mu_1 - \mu_2 \Vert^2 + \text{tr}( {\Sigma_1}) + \text{tr}( {\Sigma_2})- 2\text{tr}( (\Sigma_1^{\frac{1}{2}} {\Sigma_2} \Sigma_1^{\frac{1}{2}})^{\frac{1}{2}})}  = \sqrt{\Vert \mu_1 - \mu_2 \Vert^2 + \text{tr}( {\Sigma_1}) + \text{tr}( {\Sigma_2})- 2\text{tr}( (\Sigma_1 \Sigma_2)^{\frac{1}{2}})}
> $$
>
> We sincerely appreciate the time and effort each reader has dedicated to reviewing our work. We strongly encourage readers to provide their comments and offer their perspectives on the **objectivity and impartiality**.
>
> Best Regards, 1683 Authors

---

### Meta-Review · Area_Chair_WGoX · 2024-12-21

**Metareview:**

This paper proposes a vector quantization (VQ) method to deal with the  training instability and codebook collapse issues. The main idea is to introduce a distribution-matching approach based on the Wasserstein distance.

Strengths:
1. Novelty: The use of Wasserstein distance  (of course, under Gauss assumption) for VQ optimization is an interesting idea for aligning feature and code vector distributions.
2. Theoretical Analysis: The authors presented some mathematical analysis demonstrating improvements.

Weaknesses:

1. The strong Gaussian distribution assumption might be impractical for high-dimensional data in practical applications, as noted by multiple reviewers. The authors’ rebuttal did not adequately address this concern.

I would like to add  a few more comments on Gaussian assumption since the authors mentioned diffusion models. In the original work of diffusion models, in fact the reverse Gauss distribution assumption is asymptotically correct when the time interval approaches zero, which in fact theoretically justifies the use of Gaussian distribution in the reverse process.  In current paper. I fully understand that Gaussian might be a natural choice due to its simplicity, but more justification is needed. For example, using the illustration of Gaussian setting, the authors argued that their method works independently of codebook initialization. How can this be justified for non-Gaussian features using Wasserstein distance as an approximation?


2. The novelty is marginal. The authors’ rebuttal, while clarifying some distinctions, did not convincingly establish the novelty of their method.

Regarding the critics of reviewer YHdN on prior works of optimality of VQ, I think his/her main concern was that what is the main difference between the previous well-studied traditional rate-distortion perspective of optimality and the newly stated optimality in current paper? For down-stream applications, what fundamental differences do the two different views lead to, and why should there are such differences?

3. The method provides some marginal gains in reconstruction quality compared to previous methods.

Overall, while the paper introduces a very interesting perspective, but the weaknesses mentioned above make the contributions insufficient for acceptance at this stage. The authors’ responses did not fully resolve the reviewers' concerns  about the practical benefits and robustness of their method. I have to recommend rejection.  The decision does not  indicate the idea itself but more on the improvement to be done to make it sufficient to get published.

**Additional Comments On Reviewer Discussion:**

The reviewers raised several critical concerns during the discussion.

For example, several reviewers (e.g., YHdN, bT9x) criticized the assumption of Gaussian for feature vectors, which is unrealistic for high-dimensional data, Several reviewers (e.g., 9xEZ) questioned the novelty of the approach given previous works.  The authors provided some justification based on mathematical convenience and other validations but the reviewers did not  found them convincing enough.

Moreover, in their rebuttal, the authors argued that "Concurrently, the central limit theorem (CLT) plays a role in ensuring that learned feature vectors and code vectors follow a Gaussian distribution provided a large sample size and a large codebook size. " in the response Reviewer bT9x. While CLT is a solid and beautiful result, it is unclear how this can justify the authors' statement.   Reviewer bT9x also asked what are the distributions of the feature vectors, but they did not response.

---

### Decision · Program_Chairs · 2025-01-22

Reject